geology

DEM, plate collision, palaeolakes, Jiacha Gorge, watershed, flow reverse

**Author for correspondence:**
Yong Liu
e-mail: 1039786137@qq.com

# River capture in the middle reaches of the palaeo-Yarlung Zangbo River

Yong Liu[1,2], Yunsheng Wang[1], Liangshuai Wei[2],
Tong Shen[1], Qinfeng Shu[2], Anbang Huang[2] and Yi Jia[2]

[1]State Key Laboratory of Geo-Hazard Prevention and Geo-Environment Protection, Chengdu University of Technology, Chengdu 610059, People's Republic of China
[2]Institute of Exploration Technology, Chinese Academy of Geological Sciences, Chengdu 710043, People's Republic of China

YL, 0000-0003-2018-0144; YW, 0000-0002-1774-9494

There are 51 tributaries in the middle reaches of the Yarlung Zangbo River (YZR), and the confluences of 87% of the tributaries west of Jiacha Gorge are high-angle or perpendicular, reflecting the anomalous development of these tributaries. In this paper, field investigation and digital elevation model (DEM) methods were used to analyse the causes of this anomalous phenomenon, and it was found that there was a watershed in the area of the Jiacha Gorge. The palaeo-YZR west of the Jiacha Gorge flowed westward before the early Pleistocene into the Zada, Zhongba, Jilong and Gamba–Dingri palaeolakes, which featured a large amount of total accommodation space in the western Qinghai–Tibet Plateau; thus, this river was a continental river. With the intensification of the collision between the Indian plate and the Eurasian plate, the Qinghai–Tibet Plateau experienced rapid uplift and formed a landscape with high elevations in the west and lower elevations in the east, promoting the headward erosion of the eastward-flowing river. During the early Pleistocene, the river east of the Jiacha Gorge crossed the watershed and captured the palaeo-YZR, causing a reversal in the flow direction of the palaeo-YZR.

## 1. Introduction

The Yarlung Zangbo River (YZR) is the highest river in the world. It originates from the Majieyangzom glacier in the southwest of the Tibetan Plateau and flows from west to east. The YZR flows southward around Namjag Barwa, the eastern end of the Himalayas, and finally flows into the Indian Ocean. In recent years, river capture in the YZR Basin has attracted the attention of many scholars. A large number of research results show that the YZR formed after the early convergence of rivers between the Gangdese Mountains and the Himalayas in response to

**Figure 1.** Location of the Yarlung Zangbo River on the Qinghai–Tibet Plateau.

crustal uplift [1–7]. Wang *et al.* [1] studied the climate and topography of the lower reaches of the YZR and concluded that the Brahmaputra River, located in the tropical monsoon belt of the Indian Ocean, extended northward and captured the YZR, resulting in a sudden southward change in the YZR downstream. Chen *et al.* [8] believe that the Grand Gorge of the Yarlung Zangbo once belonged to the Palong Zangbo River system. At approximately 30 ka, the Palong Zangbo River captured the palaeo-YZR through headward erosion. Through fieldwork, Bracciali *et al.* [9] found that the early Miocene sediments in the Surma Basin on the Indian plate originated mainly from the Asian plate and was transported by the rivers in the eastern Himalayas that migrated to the north and captured the YZR. Vance *et al.* [10] analysed digital elevation models (DEMs) from the Advanced Spaceborne Thermal Emission and Reflection Radiometer (ASTER) (30 m) and the Shuttle Radar Topography Mission (SRTM) (90 m) and found that there were some signs of river capture in the lower reaches of the Himalayas. Based on remote sensing analysis, geological survey and chronological study, Shi *et al.* [11] concluded that the nearly east–west-trending palaeo-YZR developed in the Pliocene and captured the nearly north–south-trending rivers that developed in the Eocene–Miocene. The north–south-trending rivers became the tributaries of the YZR, forming the present water system. The main tributaries of the YZR, such as the Nianchu River, Lhasa River and Yalong River, flow from east to west. Based on this, Burrard & Hayden [12] suggested that the flow direction of the YZR was also from east to west before integration, contrary to the current flow direction. Li [13] considered that with the elevation difference between the west and east (high in the west and low in the east) of the YZR Basin developed gradually during the uplift of the Qinghai–Tibet Plateau and that during this process, the headward erosion of the eastward-flowing river intensified and was conducive to river capture.

However, based on field investigation data and geological structure analysis, Yang [14] concluded that the sudden change in flow direction southward in the lower reaches of the YZR was caused by structural faults rather than by river capture.

At present, many studies emphasize river capture in the lower reaches of the YZR, while river capture in the middle reaches of the YZR is rarely reported. There is a lack of systematic and comprehensive studies on river capture in the middle reaches of the YZR, but it is of great significance for the study of the tectonic evolution process in the YZR Basin. In this paper, the process of river capture and its evolution in the middle reaches of the palaeo-YZR were analysed by geomorphologic investigations in the field, DEM analysis and comprehensive data analysis.

## 2. Study area and geological background

The YZR has an average elevation higher than 4000 m, a total length of 2057 km and a catchment area in China of 240 480 km². The middle reaches of the YZR extend from Zhongba County in the west (30°13′ N, 83°14′ E) to the town of Pai in the east (29°30′ N, 94°52′ E), with a distance of approximately 1184 km (figure 1). Wide valleys and gorges alternate in the river basin. According to the topographic

**Table 1.** Information of the global DEMs and reference SRTM version. DEM data are from http://www.gscloud.cn/.

| SRTM version | digital elevation data | band | special resolution (m) | central longitude (°) | central latitude (°) | horizontal/ vertical datum |
|---|---|---|---|---|---|---|
| SRTM3 | srtm_53_06 | C band | 90 | 82.5 | 32.5 | WGS84/EGM96 |
| SRTM3 | srtm_53_07 | C band | 90 | 82.5 | 27.5 | WGS84/EGM96 |
| SRTM3 | srtm_54_06 | C band | 90 | 87.5 | 32.5 | WGS84/EGM96 |
| SRTM3 | srtm_54_07 | C band | 90 | 87.5 | 27.5 | WGS84/EGM96 |
| SRTM3 | srtm_55_06 | C band | 90 | 92.5 | 32.5 | WGS84/EGM96 |
| SRTM3 | srtm_55_07 | C band | 90 | 92.5 | 27.5 | WGS84/EGM96 |

characteristics of the river valley, the river basin can be divided into four wide valleys, namely, the Milin, Shannan, Shigatse and Maquan river valleys [15]. The field investigation showed that the river in the wide valleys has a slow flow and many tributaries. The width of the river surface is approximately 200–400 m, with a maximum width of 2 km. The width of the valley bottom is 2–4 km, with a maximum width of more than 7 km. The wide valleys are alluvial river channels. In the dry season, a large amount of alluvial material is exposed above the water. The gorges between the wide valleys are generally narrow and deep. The width of the river surface is 50–150 m, and the width of the bottom of the valley is generally 100–260 m. In the gorges, the elevation of the river drops greatly, and the flow rate is very high. Thus, the bedrock gorges are areas of high erosion rates and stream power. The Jiacha Gorge is the deepest gorge in the middle reaches of the YZR.

The river is located along the east–west-trending fault zone between the Himalayas and the Gangdese Mountains. This zone, known as the Yarlung Zangbo Suture Zone, was formed by the collision between the Eurasian plate and the Indian plate [16]. The initial stage of collision between the Eurasian plate and Indian plate occurred in the early Palaeocene (approx. 65 Ma), the full collision stage occurred at the end of the Palaeocene (approx. 55 Ma), and the continuous collision stage occurred at the end of the Eocene (40–38 Ma) [17]. The initial collision and full collision stages were mainly associated with the protruding part of the frontal edge of the Indian plate. The two plates sutured completely during the continuous collision stage, and the Tethys Ocean between the two plates disappeared at this time [18]. The three stages of collision uplifted the crust and formed the initial Qinghai–Tibet Plateau [19]. In the late Eocene, the initial Qinghai–Tibet Plateau began to experience intense convergence with respect to areas farther inland, causing the plateau to rise rapidly and the relative tectonic processes, such as subduction, nappe formation, faulting and extension, to appear at this time [20].

# 3. Material and methods

## 3.1. Digital elevation models and pre-processing of the river network

The DEMs used in this study were derived from SRTM3 data. At present, the SRTM3 data available for China have a resolution of 3 arc-sec, i.e. 90 m; similar-resolution data cover the whole world. The elevation reference system of SRTM3 is the geodetic datum of EGM96, and the horizontal reference system is WGS84. The nominal absolute elevation accuracy is ±16 m, and the absolute horizontal accuracy is ±20 m [21]. All digital elevation data for this study were collected from the China Geospatial Data Cloud (CGDC). Further details on the SRTM data are listed in table 1.

ArcGIS 10.2 software was used to read the acquired digital elevation data and generate the DEM. The DEM was pre-processed using the hydrological calculation module in the software to extract the river network in the middle reaches of the YZR, and the specific process is shown in figure 2. First, the AGREE method is used to smooth the DEM so that the processed DEM is consistent with the input vector elevation data; then, the DEM depressions were filled by calculating the depth and contribution rate of the depressions and setting a reasonable filling threshold of 6 m in conjunction with the real geomorphological conditions. Depressions less than the threshold were filled, and depressions greater than the value were retained. The D8 method is used to calculate the flow direction of the river, that is, to calculate the steepest direction between each grid and its adjacent grid. Then, the catchment area of the upper reaches of the grid is calculated to extract the river network, which is shown in figure 3a

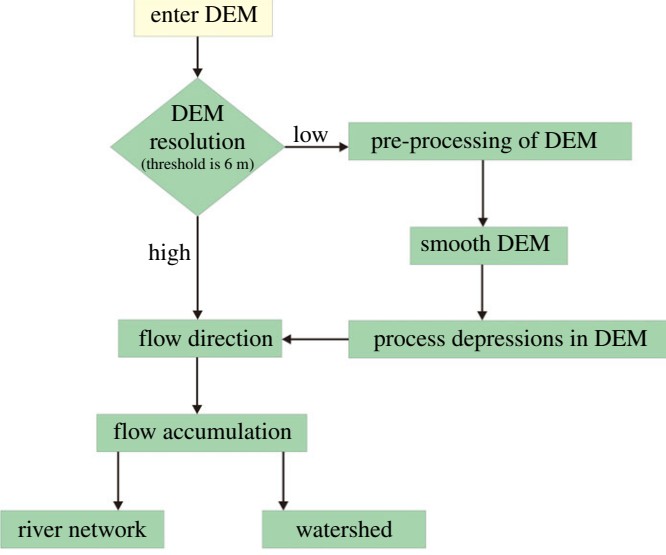

**Figure 2.** Flow chart of the river network extraction.

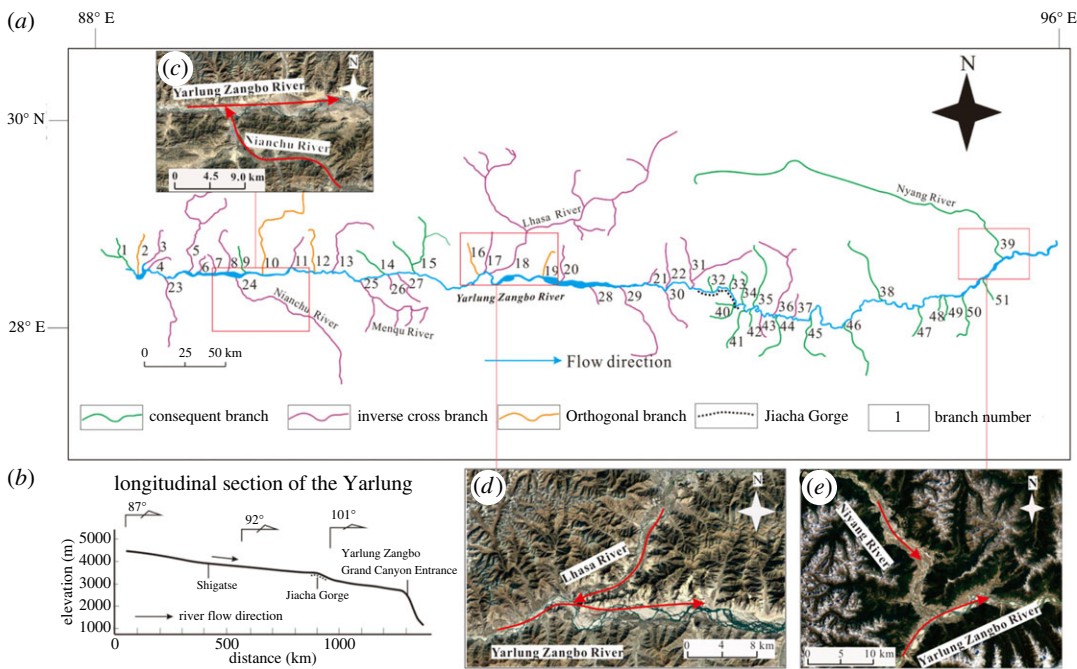

**Figure 3.** (*a*) River network in the middle reaches of the YZR. (*b*) Longitudinal section of Yarlung Zangbo River. (*c*) Intersection of Nianchu River and Yarlung Zangbo River. (*d*) Intersection of Lhasa River and Yarlung Zangbo River. (*e*) Intersection of Niyang River and Yarlung Zangbo River.

(the catchment area threshold is set to 6 km$^2$). Based on the elevation of the DEM, a river profile of the middle reaches of the YZR is shown in figure 3*b*.

# 4. Results

## 4.1 Characteristics of the river network

The river network map of the middle reaches of the YZR (figure 3*a*) shows that the YZR and its tributaries have been substantially deformed by plate convergence, and the whole river network is narrow in the north–south direction and wide in the east–west direction. The directions of the primary tributaries

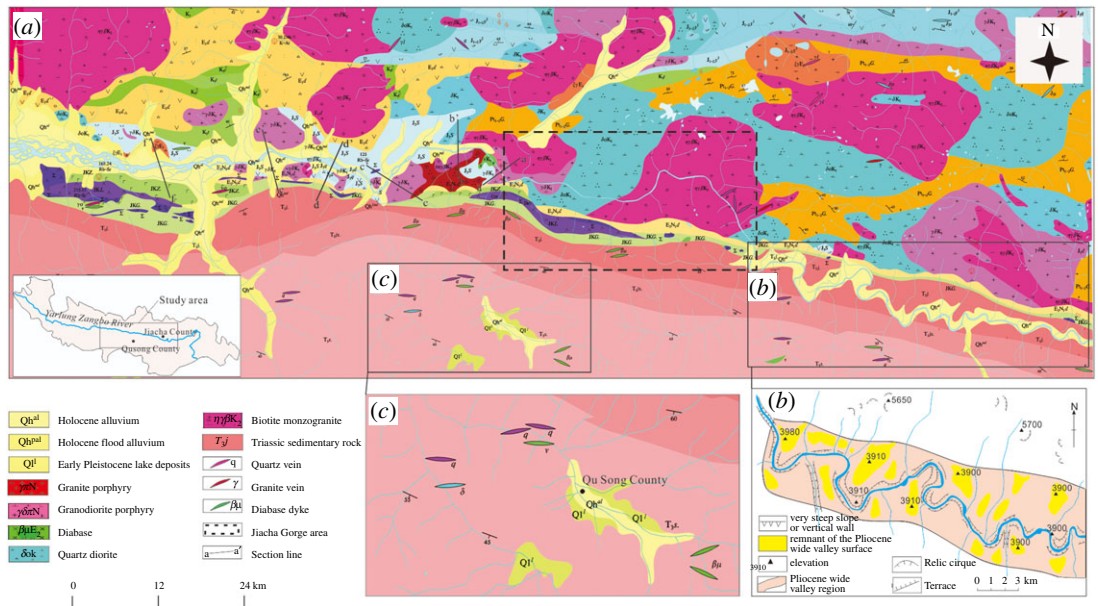

**Figure 4.** Tectonic geological map of the middle reaches of the YZR. (*a*) is the tectonic geological map from Qushui County to Jiacha County; (*b*) is the meandering river located at the lower reaches of the Jiacha Gorge outlet; (*c*) is the Quaternary sediments in Qusong Basin.

**Table 2.** Direction of the branches flowing into the YZR.

| river section | riverbank | direction of the branches flowing into the YZR | | |
| --- | --- | --- | --- | --- |
| | | consequent | orthogonal | inverse |
| West of Jiacha Gorge | left bank | 4 | 5 | 14 |
| | right bank | 0 | 0 | 8 |
| East of Jiacha Gorge | left bank | 6 | 0 | 2 |
| | right bank | 11 | 0 | 1 |

flowing into the YZR are not the same. For example, the Nianchu River and Lhasa River flow into the YZR at a high angle to the flow direction of the YZR, while the Niyang River flows into the YZR at a low angle (figure 3*c–e*). The directions of the tributaries flowing into the YZR are listed in table 2. There are a total of 31 tributaries of the YZR located west of Jiacha Gorge. On the left bank, 14 tributaries flow into the YZR at high angles, five tributaries flow into the YZR perpendicularly and the other four tributaries flow into the YZR at low angles. All eight tributaries on the right bank flow into the YZR at high angles. To the east of Jiacha Gorge, there are a total of 20 tributaries. Among them, six tributaries on the left bank flow into the YZR at low angles, and two tributaries flow into the YZR at high angles. On the right bank, 11 tributaries flow into the YZR at low angles, and one tributary flows into the YZR at a high angle. A total of 87% of tributaries west of Jiacha Gorge are high-angle and perpendicular rivers, whereas 85% of the tributaries east of Jiacha Gorge are low-angle rivers.

From the river profile (figure 3*b*), the elevation of the riverbed gradually decreases from 4500 m in the west to 1020 m in the east. From Zhongba County to Jiacha Gorge, the average longitudinal gradient of the riverbed is 0.7‰, and the average longitudinal gradient of the riverbed from the Jiacha Gorge to the town of Pai is 2.5‰. The elevation of the riverbed in the Grand Gorge of the Yarlung Zangbo drops sharply, and the longitudinal gradient of the riverbed can reach up to 10.3‰.

Six cross-sectional maps of the YZR valley west of Jiacha Gorge, i.e. sections a–a′ to f–f′, were constructed on the basis of the DEM and field investigation data (figures 4 and 5). From the sections, the third-level planation surfaces range in elevation from 3857 to 4337 m, and the average elevation of the planation surfaces on the left bank of the YZR is 63 m higher than that on the right bank. From the Jiacha Gorge (a–a′ section) to Sangri County (c–c′ section), the elevation of the third-level

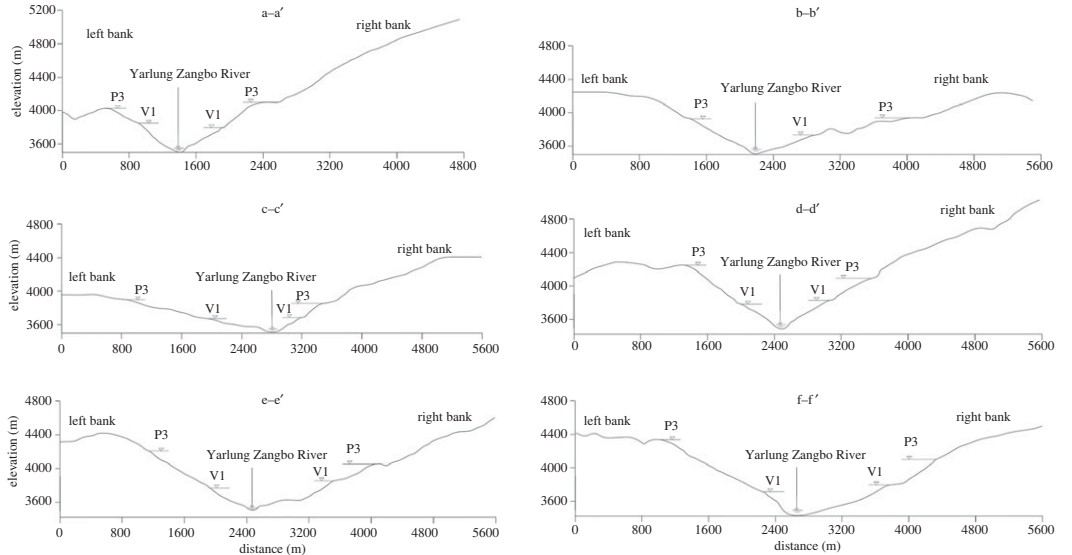

**Figure 5.** Transverse section of the YZR west of Jiacha Gorge. P3 is the third-level planation surface, and V1 is the first-grade valley shoulder.

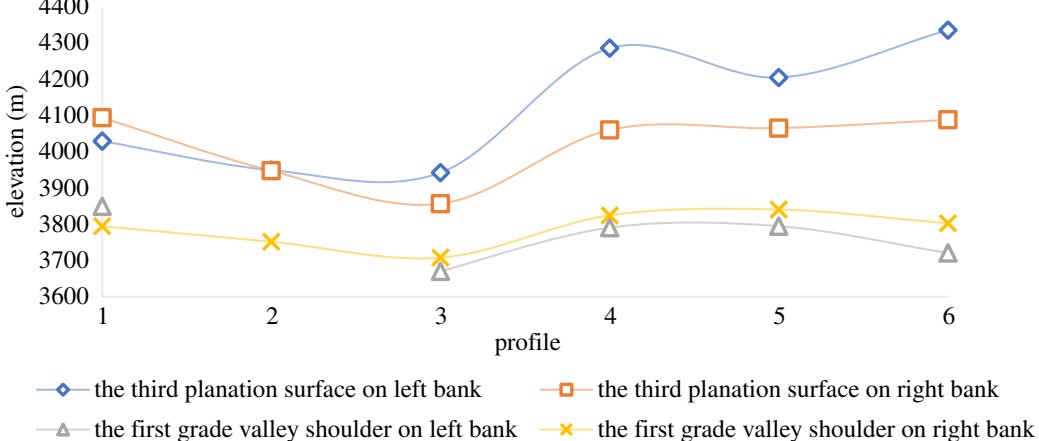

**Figure 6.** Planation surface and valley shoulder elevation in different parts of the middle reaches of YZR. b–b′ profile missing a first-grade valley shoulder on the left bank.

planation surfaces gradually decreases, while from Sangri County (c–c′ section) to the Zedang area (f–f′ section), the elevation of the third-level planation surfaces gradually increases. The first-stage valley shoulders range in elevation from 3670 to 3841 m, and the average elevation of the first-stage valley shoulders on the north bank is 22 m lower than that on the south bank. From the Jiacha Gorge to the west, the elevation trend of the first-stage valley shoulder is consistent with the trend of the third-level planation surfaces, exhibiting a 'V'-shaped trend of first decreasing and then increasing. The elevations of the third-level planation surfaces and the first-stage valley shoulders are lowest in Sangri County (figure 6). The first-stage valley shoulder on the left bank of the Luobusha area (b–b′ section) has been eroded, and those in Rong County (d–d′ section and e–e′ section) were found to be very small. From the 1:250 000 regional geological survey of the Zedang area (Geological Survey Institute of Tibet, 2007), the third-level planation surface of the YZR was formed in the late Pleistocene.

The Jiacha Gorge and its downstream reaches are dominated by meandering rivers. The river channel of the Jiacha Gorge section is characterized by a steep gradient, a high flow speed and a strong ability to transport river material. The river water has strongly eroded and cut the riverbed, resulting in the formation of a deep 'V'-shaped valley. In this bedrock setting, deposits are not expected in the reach due to the high stream power and high transport capacity of the water [22]. The bedrock on either side of the river valley is composed of Cretaceous granite, with a hard texture and a strong resistance to weathering and erosion, and the slopes of the two sides are greater than 50°.

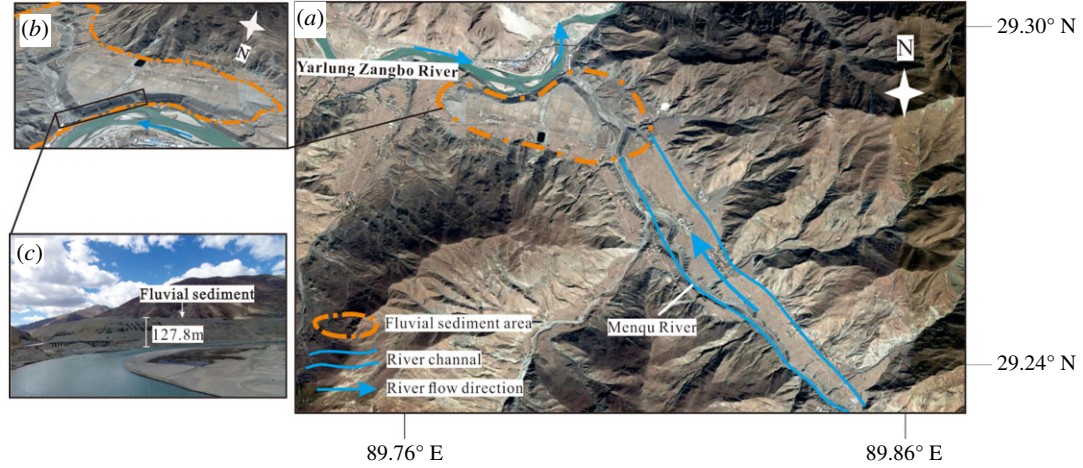

**Figure 7.** (*a*) Flow direction and fluvial sediments of the Menqu River. (*b*) Relationship between fluvial sediments and the outlet of Menqu River. (*c*) Thicknesses of the alluvial sediments.

## 4.2. Sedimentary characteristics of the tributary deposits

The alluvial sediments transported by the tributaries are affected by the YZR during their deposition process at the tributary outlets, and the deposits reflect the hydrodynamic characteristics of the YZR. To clarify the hydrodynamic characteristics of the YZR, we conducted field investigations in typical tributaries, such as the Menqu River and the Nianchu River. From the geological map (figure 4), the north bank of the YZR is mainly composed of granite, diorite and some other magmatic rocks, while the south bank is mainly composed of sedimentary rocks formed in the Triassic period. The Nianchu River and the Menqu River, which are located on the right bank of the YZR, are both developed in sedimentary areas.

The Menqu River is a high-angle tributary of the YZR with a length of 73 km and flows into the YZR from southeast to northwest (figure 7*a*). The alluvial sediments transported by the Menqu River are mainly deposited on the west side of the outlet, and the thicknesses of the alluvial sediments are found to be large, i.e. 127.8 m (figure 7*b*,*c*). The alluvial sediments deposited at the outlet and terraces of the Menqu River are made of cobbles and pebbles with intermittent sand beds and lenses and are grey-black in colour, the intermittent sand beds of 10–40 cm thickness were deposited 3.5 m below the surface, and the lens observed in the alluvial sediments of the Menqu River are made of well-imbricated subangular clasts, which are inclined to the east (figure 8*a*).

Similar to the Menqu River, the Nianchu River is also a high-angle tributary and flows into the YZR from southeast to northwest (figure 9*a*). From the field investigation data, the sediments on the west side of the Nianchu River outlet are much higher than those on the east side, and an approximately 2.4 m high scarp appears in the middle due to the sudden change in the sediment thickness (figure 9*b*).

## 4.3. Palaeolakes on the western Qinghai–Tibet Plateau

In recent years, several palaeolakes have been identified on the western Qinghai–Tibet Plateau, such as the Zhada, Zhongba, Jilong, Dati and Gangba–Dingri Palaeolakes. Among them, the Zhada and Zhongba Palaeolakes were located west of the YZR suture zone, whereas the Jilong, Dati and Gangba–Dingri Palaeolakes were located southwest of the YZR suture zone and north of the Himalayas. These palaeolakes formed at 7.0–1.36 Ma and experienced the evolutionary process of formation, expansion, shrinkage and disappearance [23–29].

The Zhada Palaeolake (30°50′–32°20′ N, 79°00′–80°30′ E) was elongated in the northwest–southeast direction and was long and narrow, with a width of approximately 70 km and a length of approximately 260 km. The lake basin basement is composed of pre-Jurassic sandstone and limestone, which is covered by fluvial–lacustrine sediments with a thickness of nearly 1000 m; the area of the sediments is as large as 9000 km$^2$ [27,30]. These characteristics indicate that the estimated water capacity of the palaeolake was $9 \times 10^{12}$ m$^3$. Saylor *et al.* [31] found that the sedimentary profiles of the Zhada Palaeolake include fluvial-supratidal, littoral, fluvial and delta sediments from bottom to top. This finding reflects a complete evolutionary process from formation to expansion to shrinkage

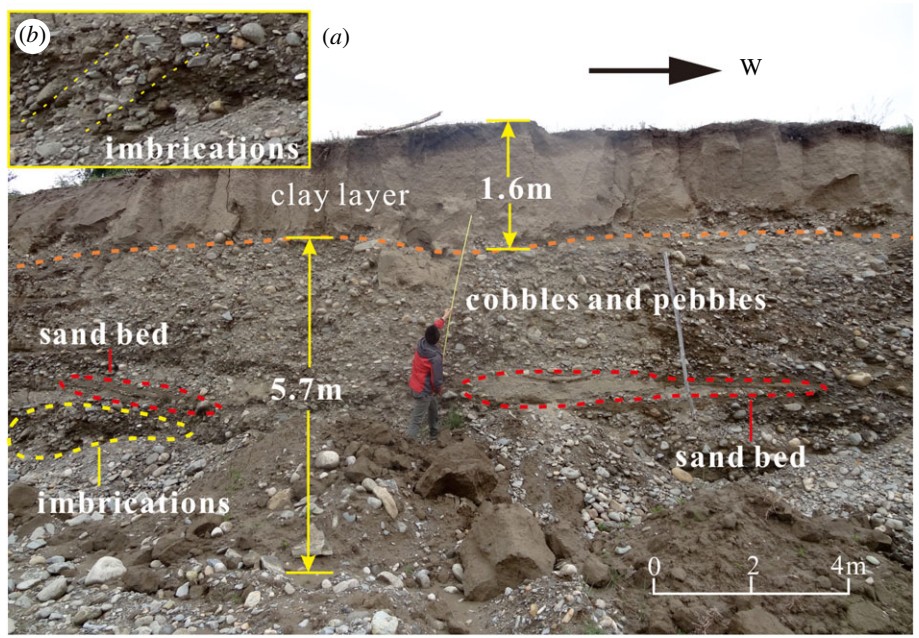

**Figure 8.** The alluvial sediments characteristics of the Menqu River. (*a*) Sediment profile of the Menqu River, (*b*) Characteristics of the imbrications.

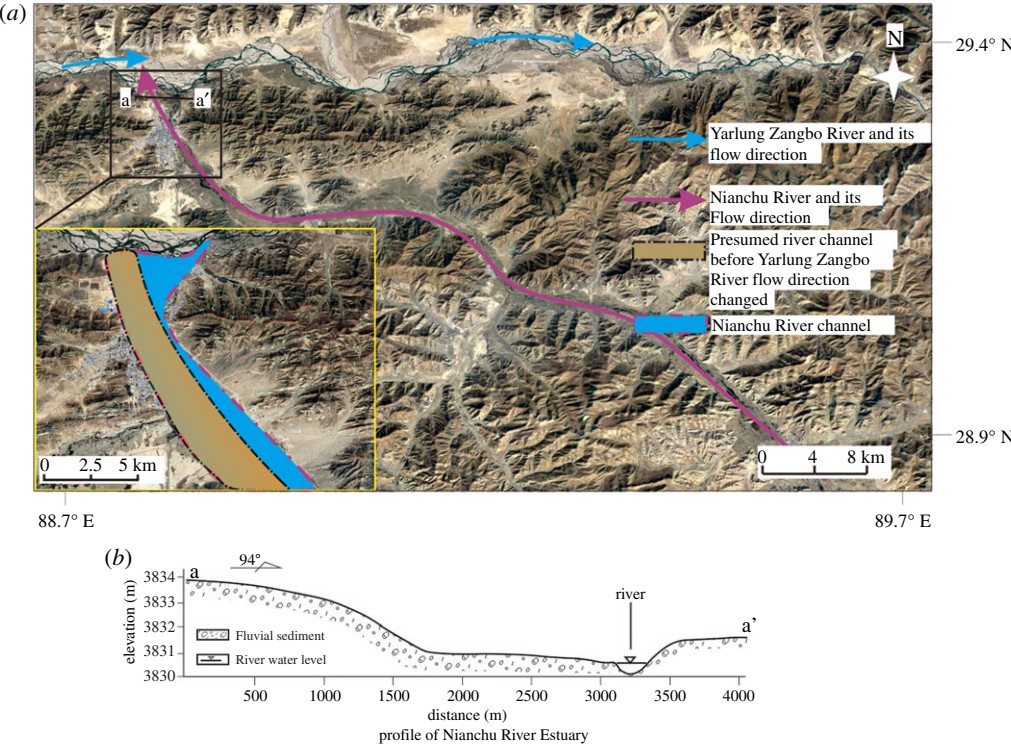

**Figure 9.** (*a*) Flow direction and fluvial sediments of the Nianchu River. (*b*) Profile of the Nianchu River Estuary.

[32,33]. Zhu *et al*. [26] measured the age of the palaeolake sediments and found that the Zhada Palaeolake began to form at approximately 5.4 Ma and began to shrink at approximately 3.2 Ma.

The Zhongba Palaeolake (29°21′–30°12′ N, 82°18′–83°24′ E) was elongated in the northwest–southeast direction and had a width of approximately 34 km and a length of approximately 126 km. Additionally, the palaeolake basin covered an area of 4284 km². Liu *et al*. [23] acquired a lacustrine sediment profile to a depth of 43 m in the palaeolake, and through sampling and dating of the

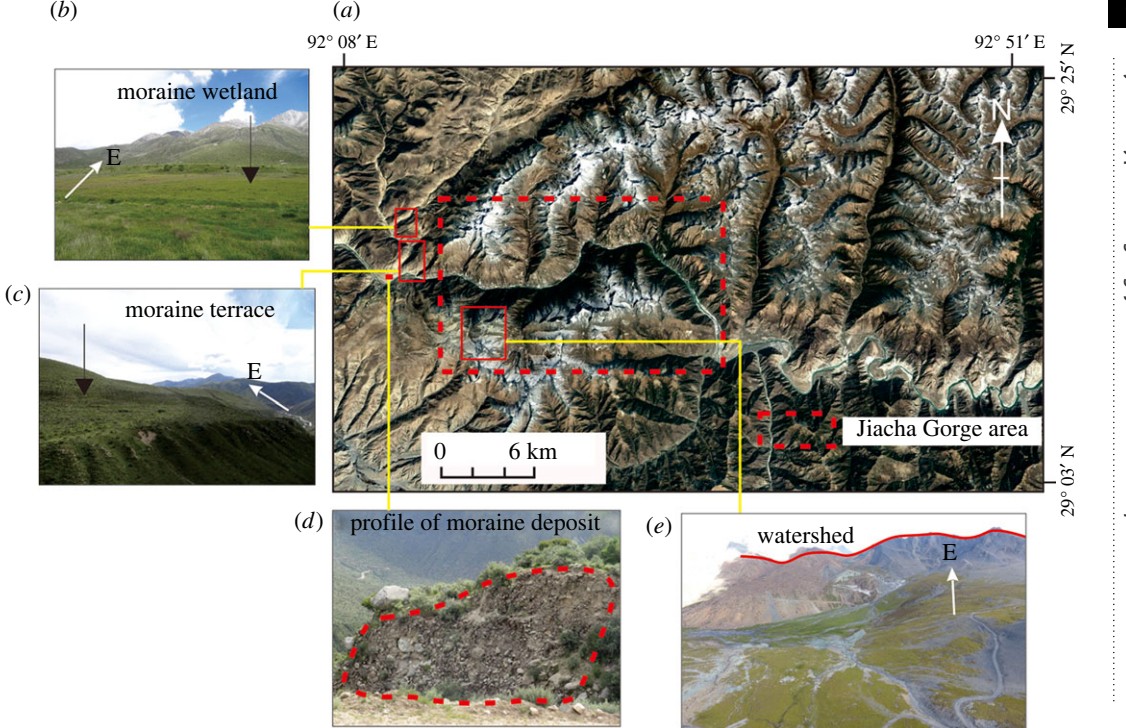

**Figure 10.** Moraine terraces on the west side of Jiacha Gorge. (*a*) Spatial distribution map of the moraine deposits and watershed. (*b*) Characteristics of the moraine wetland. (*c*) Characteristics of the moraine terrace. (*d*) Profile of moraine deposit. (*e*) Characteristics of the watershed.

sediments, they found that the palaeolake began to form at approximately 4 Ma and reached its maximum area and began to shrink at approximately 3.45 Ma.

The Jilong Palaeolake Basin is located on the southern bank of the YZR (28°18′–29°17′ N, 84°29′–86°05′ E) and is narrow in the north and wide in the south, with an area of approximately 300 km². Wang & Shen [25] carried out a field investigation on the palaeolake and found that the thickness of the lacustrine sediments reaches a maximum of 300 m. There are some alluvial sediments deposited in the upper part of the lacustrine sediment profile. These alluvial sediments indicate that the water in the palaeolake flowed out of the system, and during this period, the palaeolake began to shrink. Wang & Shen [25] found that the formation times of the bottom sediments and the alluvial sediments in the lacustrine sedimentary profile were approximately 7 Ma and 3.4 Ma, respectively. That is, the palaeolake formed at approximately 7 Ma and began to shrink at 3.4 Ma.

The Gangba–Dingri Basin is located in the northern foot of the Himalayas and is adjacent to the YZR suture zone (28°12′–28°48′ N, 87°05′–88°03′ E). This basin is approximately 900 km from east to west and approximately 50–70 km wide from north to south, with an area of approximately 70 000 km². Through field investigation, Du *et al.* [29] found that a large number of thick shales were deposited in the basin and concluded that a palaeolake had once existed. The Miocene Gangba–Dongshan Formation is widely distributed, mostly in Gangba and Dingri counties. This formation is composed of grey-black thin-layered shale and silty shale. These shale layers contain fossils such as ammonites and brachiopods and are 190–1100 m thick. These characteristics indicate that the Gangba–Dingri Palaeolake had a deep water depth, a large area, and a large water capacity.

In summary, the palaeolakes distributed in the western Qinghai–Tibet Plateau reached their greatest extents and began to shrink at 3.2–3.45 Ma, and they finally disappeared at approximately 1.7 Ma. The deposits that formed in these palaeolakes have a wide distribution area and a large sediment thickness, indicating that the total accommodation space was high.

## 4.4. Geomorphological characteristics of the Jiacha Gorge

The field investigation revealed that there are many moraine terraces located on the left bank of the Woka River, which is located on the west side of Jiacha Gorge (figure 10*a–d*). The surfaces of these terraces are

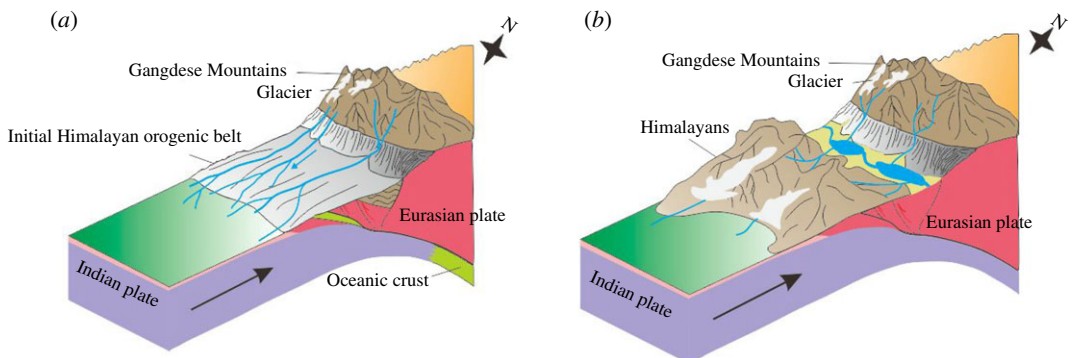

**Figure 11.** Evolution map of the YZR. Black arrow points to the direction of the plate movement. (*a*) Characteristics of water system in the early stage of the uplift of the Gangdese Mountains. (*b*) Characteristics of water system after Himalayas uplift.

relatively flat and are inclined to the west. The thickness of the moraine terraces is 80–200 m. Eroded by the Woka River, many steep slopes of 45–70° have formed on the front edge of the terraces. The moraine terraces were formed by the westward movement of glaciers along the eastern side of the Jiacha Gorge, and the sediments are poorly sorted and have good water permeability. The Woka River and the adjacent Zengqi River originate from glaciers on the eastern side of the Jiacha Gorge.

On the southern bank of the Jiacha Gorge, a watershed is observed in the Kangjinla area at an elevation of 5400 m (figure 10*e*). The watershed extends from south to north and disappears at the Jiacha Gorge. The rock strata of the watershed dip to the west at an angle of more than 45°. Due to the high elevation, low temperature and low rainfall in the Kangjinla area, the watershed features little vegetation cover. The vegetation in the Jiacha Gorge and the region downstream to the east of the watershed is mostly composed of small trees, while the vegetation to the west of the watershed is poorly developed and features low shrubs.

# 5. Discussion

## 5.1. Analysis of anomalous deposits at the tributary outlets

Based on hydrodynamics, the alluvial sediments carried by tributary rivers are transported and deposited downstream after entering the main trunk river. Therefore, the alluvial sediments originating from a tributary river should be mainly deposited on the eastern side of the tributary outlet based on the hydrodynamic characteristics of the YZR. However, the hydrodynamic characteristics of the alluvial sediments of some tributary rivers, such as the Menqu River and the Nianchu River, are inconsistent with those of the YZR. From the field investigation, there are large amounts of alluvial sediments deposited on the western sides of the Menqu River and the Nianchu River outlets, with only small deposits on the eastern sides. The well-imbricated subangular clasts in the lens observed in the sediments of the Menqu River are inclined to the east, which shows the sediments once affected by the westward flow in the sedimentation process (figure 8*b*). These results indicate that the YZR probably flowed westward for a long period in the past. When the YZR flowed westward, the sediments carried by the tributary rivers were deposited on the western side of the tributary outlets, forming a large sedimentary platform under these long-term hydrodynamic forces. Later, when the flow direction of the YZR reversed towards the east, the sediments of the tributary rivers began to be deposited on the eastern side of the tributary outlets.

## 5.2. Analysis of river network formation mechanism

The YZR is located between the Himalayas and the Gangdese Mountains. In the process of the continuous collision between the Indian plate and the Eurasian plate, the Gangdese Mountains rose before the Himalayas [1]. The Gangdese Mountains started to rise in the late Mesozoic, and many rivers originating from the Gangdese Mountains and the northern Tibetan Plateau first flowed southward into the Tethys Sea and then flowed to the Indian continent after the closure of the Tethys Sea (figure 11*a*). In the early Miocene, the Himalayas began to rise and blocked the rivers flowing south, resulting in the convergence of the river water between the two orogenic belts of the Himalayas and the Gangdese Mountains [34]. The collision between the Indian plate and the Eurasian plate first occurred in the middle of the YZR collision zone and gradually extended to the east and west, and spatial differences in

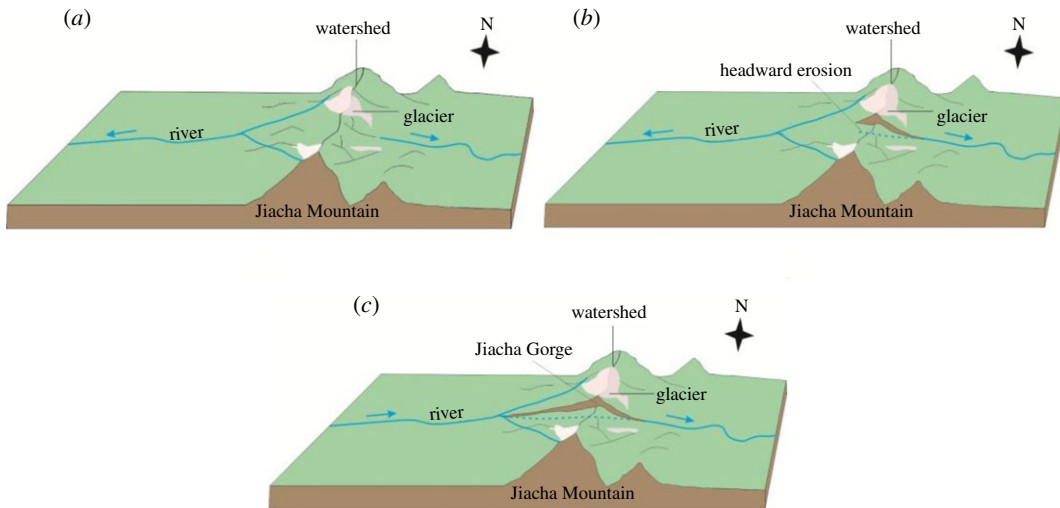

**Figure 12.** Evolution map of the Jiacha Gorge. (*a*) Water system before the Jiacha gorge was formed. (*b*) The watershed of Jiacha Mountain was destroyed by the headward erosion of river. (*c*) Water system after the Jiacha gorge was formed.

the collision resulted in the YZR suture zone consisting of ridges and troughs [35]. The river water from the Himalayas and the Gangdese Mountains converged into the troughs to form lakes. Before the end of the late Tertiary period, the elevation of the Qinghai–Tibet Plateau was less than 2000 m, and the Indian Ocean monsoon was able to transport a large amount of moisture to Tibet, resulting in abundant rainfall; thus, the rivers between the two mountains had high flow rates and strong erosional forces [36,37]. After the erosion of ridges by river water and the rainfall between the troughs, the lakes became connected to form an early version of the YZR (figure 11*b*) [24,38].

Most tributaries west of the Jiacha Gorge are high-angle and perpendicular rivers. From the field investigation data, we do not think that the abnormal relationship between the tributaries and the YZR was caused by tectonism. If the left bank tributaries of the YZR flowed west due to westward tectonic action, the right bank of the YZR should exhibit eastward tectonism, resulting in the right bank tributaries of the YZR flowing east. However, most of the tributaries on both banks flow westward or are perpendicular to the YZR, and strike-slip faults that could potentially influence the flow direction of the tributaries were not found in the field investigation. The sediments deposited at the outlets of some tributaries of the YZR indicate that the YZR flowed westward over a long period in the past. When the YZR flowed westward in response to topography, its tributaries would have also flowed westward before merging with the YZR.

One watershed is observed in the Jiacha Mountain, indicating that the western and eastern sides of the watershed once belonged to different hydrologic systems. Later, after the Jiacha Gorge formed, the watershed was destroyed, and the two river systems merged. With a length of approximately 50 km, the Jiacha Gorge is the deepest gorge in the middle reaches of the YZR, and is composed of Cretaceous granite. The high-angle and perpendicular tributaries of the YZR are mostly located to the west of the Jiacha Gorge. Therefore, the westward flow of the YZR mainly occurred in the valley west of the Jiacha Gorge. Since there was no outflow river and fluvial sediments are observed on the western side of the YZR suture zone, the westward-flowing river was probably an endorheic basin. The palaeolakes, with a large amount of total accommodation space, on the western side of the YZR were probably the final destination of the early westward-flowing YZR.

## 5.3. Formation time of the Jiacha Gorge

At approximately 3.4 Ma, due to the continuous collision between the Indian plate and the Eurasian plate, the Tibetan Plateau began to rise rapidly, and the terrain difference between the high-elevation western portion and the low-elevation eastern portion of the Tibetan Plateau gradually increased [39,40]. The rapid uplift of the Tibetan Plateau blocked the Indian monsoon, and only a small amount of the Indian monsoon could flow upstream along the river valleys in the southeastern portion of the Tibetan Plateau, resulting in a sharp decrease in rainfall in the western portion of the Qinghai–Tibet Plateau [41]. The

rise in elevation and the sharp decrease in rainfall in the western Qinghai–Tibet Plateau are probably the reasons for the shrinkage of the palaeolakes during 3.2–3.45 Ma.

From the riverbed profile of the YZR, the average longitudinal gradient of the riverbed from Zhongba County to the Jiacha Gorge is much lower than that from the Jiacha Gorge to the town of Pai, indicating that the river east of the Jiacha Gorge flows faster than the upstream portion. To the east of the Jiacha Gorge, rainfall is abundant, and the river discharge is large, which leads to the strong headward erosion of the river. This strong headward erosion of the river formed the Jiacha Gorge, and the watershed of Jiacha Mountain was destroyed, which resulted in the capture of the early YZR west of the Jiacha Gorge (figure 12*a–c*). After capture, the flow direction of the early YZR began to reverse, leading to the disappearance of palaeolakes in the western Tibetan Plateau due to a lack of water sources. As most of the palaeolakes in the western Tibetan Plateau disappeared at approximately 1.7 Ma, the capture of the early YZR and the formation of the Jiacha Gorge probably also occurred at approximately 1.7 Ma (the early Pleistocene).

It can be observed that the Jiacha gorge formed in the study area are controlled by both climate and tectonics. Climatic control provides the essential river supply, whereas the tectonic movement provides river potential energy. The present work provides a scope for future work to do a detailed study on the crustal structure evolution of the Qinghai–Tibet Plateau.

# 6. Conclusion

This study presents valuable field investigation data of geology of the middle reaches of the YZR. DEM methods were used to analyse the characteristics of the river network. The following conclusions can be drawn from this study:

(1) Before the early Pleistocene, the palaeo-YZR west of the Jiacha Gorge flowed westward into the palaeolakes in the western Tibetan Plateau. In the context of this westward flow of the palaeo-YZR, the tributaries also flowed westward, and the sediments deposited at the outlets of these tributaries reflected the hydrodynamic characteristics of the palaeo-YZR.
(2) During the early Pleistocene, the Jiacha Gorge was formed by the strong headward erosion of the river east of the Jiacha Gorge, resulting in the capture of the palaeo-YZR west of the Jiacha Gorge. Climate and tectonics are the main controlling factors of the palaeo-YZR capture, which is consistent with the capture of the Brahmaputra River and the Palong Zangbo River in the downstream of the YZR.

Data accessibility. The graphical data on the planation surface and valley shoulder elevations in different parts of the middle reaches of the YZR and the river network flow directions that support this paper have been uploaded as electronic supplementary material.

Authors' contributions. Y.L., L.W., T.S., Q.S., A.H. and Y.J. conducted the field investigation and data analysis, Y.W. guided the writing process and Y.L. wrote the paper. All the authors gave their final approval for publication.

Competing interests. We declare we have no competing interests.

Funding. This study was supported by the Foundation of China Geological Survey (grant no. DD20190325).

Acknowledgements. We thank the editor and anonymous reviewers for their valuable comments and suggestions.

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
