## [Reviewer comments · Royal Society Open Science]

Review History

RSOS-191753.R0 (Original submission)

Review form: Reviewer 1

Is the manuscript scientifically sound in its present form?

No

Are the interpretations and conclusions justified by the results?

No

Is the language acceptable?

Yes

Do you have any ethical concerns with this paper?

No

Have you any concerns about statistical analyses in this paper?

No

Recommendation?

Major revision is needed (please make suggestions in comments)

Comments to the Author(s)

This paper combines fieldwork and DEM analysis to understand the development of the understudied middle reaches of the Yarlung Zangbo River; it combines geomorphology and sedimentology.

Currently, the evidence for the analysis is missing for some sections, especially the branch deposits. A new section in the results section is required to show the fieldwork that you undertook. The implications of your work need to be included, how does this fit with the research further downstream?

Further analysis using the DEM could be used to strengthen your analysis, such as morphometric analysis of the river systems. Normalised steepness index and hypsometry can be used to show anomalous reaches of the river systems and can be completed using GIS. It would be interesting to see if there are any relationships between the branch river types and the main trunk river.

Specific comments below:

Page 1

Line 25 – Please expand on what you mean by Quaternary investigation – do you mean your field work? Or analysing the quaternary evolution of the catchment?

Line 28 – ‘it is believed’ – is this from previous published work? if so reword to ‘previous researchers have shown’

Line 34 – Water storage capacity is not the correct term, please look at the comments for page 6 line 11.

Page 2

Section 3 – This section currently has a tectonic / structural background of the study area, geological information on the geology (bedrock and superficial including the stratigraphic successions of the palaeolake infill). Include a geological map / stratigraphic chart earlier than Figure 7. Geology is referred to throughout the paper, so setting the context here would be useful.

This section could also benefit from information on the types of river channels you are looking at – are they bedrock or bedrock-alluvial or alluvial in nature? What is the current catchment area of the river system?

Line 51 – include the elevation decrease from east to west

Page 3

Line 57 – add what filling threshold you used

Page 4

Line 5 – what area / con value did you use?

Section 5 – you refer to the deposits of the branch outlets in the analysis section (Section 6.1) – however do not present the data in the results sections from your fieldwork. What are the character of the deposits? Did you see any clast imbrication that supports flow reversal. A new sub-heading in the results section is needed to display your data to support your later interpretations.

Page 4 /5

Section 5.2 – when each lake is introduced refer to the figure number (Figure 5), it would be useful to have a schematic diagram of the ages of the lakes and thickness of deposits – do you have a sediment logs of the area?

Page 6

Line 11 – Water storage capacity is not the right term here, sedimentary thicknesses can be high in areas of subsidence and low water volumes. Total accommodation space would be a more appropriate term to use.

Line 22 / 23 – ‘Melting water from alpine glaciers...’ – I would remove this sentence, it doesn’t add anything.

Line 25 – 41 – are there any age constraints on these surfaces via dating e.g., cosmogenic or apatite fission track?

Line 25 – 41 – the focus is on the c-c’ cross sections, add in information about all the cross sections extracted (a-a’ to e-e’)

Line 43 – 58 – references on bedrock channels are missing from this section.

For example, but not limited to:

Tinkler, K.J. and Wohl, E., 1998. Rivers over rock: Fluvial processes in bedrock channels (No. 107). American Geophysical Union. And references therein

Richardson, K. and Carling, P., 2005. A typology of sculpted forms in open bedrock channels (Vol. 392). Geological Society of America.

Line 44 – Reword ‘large drop’ to ‘steep gradient’ and include the gradient of the reach

Line 48 – Reword ‘There are no quaternary deposits in the gorge due to the strong erosional downcutting of the river...’ In a bedrock setting, deposits are not expected in the reach due to the high stream powers causing high transport capacity of the water.

Page 7

Section 6 in general– what are the implications of your study? Do any of the dates correspond with information in the depositional basin? Does this support previously published research on the lower reaches or not?

Section 6.1 – the evidence is needed in the results section to support your interpretations for this section.

Section 6.2 – how does your results and interpretation fit with the other work further downstream cited in the introduction?

Page 8

Line 5 – do you have a reference for the elevation of the plateau?

Line 17 – large water capacity, see comment for page 6 line 11

Page 9

Line 14 – ‘were burst by the river’s upstream development’ reword to ‘drained by headward erosion’

Line 31 – water storage capacities, see comment for page 6, line 11.

Figures

Figure 1 – The text in the inset box should be increased, when printed, this is hard to read, a legend is needed for the DEM in the inset box.

What do the yellow dashed line refer to? Missing from the key

Figure 2 – it would be useful to include information on the thresholds you used

Figure 3 – Move the inset b, below part A – it is currently too small to see and could be made much larger if you place it below A

Figure 4 – place a white box behind the scale, and change the text to black / bold the legend – it is currently hard to see

Figure 5 – you could add the sediment logs in this figure for B, C and D (See comment for section 5.2)

Figure 7 – the red box in inset A is hard to see with the current colour scheme used

Figure 8 – are the moraines mentioned in section 5.3 visible in any of the cross sections?

Figure 14 – in the text you mention about draining of the lakes by headward erosion / shrinkage of the lakes, it would be good to show this in the schematic as section C

Figure 15 – reword caption to ‘Satellite image of the Qusong Drainage Basin’ – or add a scale for the elevation

Review form: Reviewer 2 (Ramesh Pant)

Is the manuscript scientifically sound in its present form?

Yes

Are the interpretations and conclusions justified by the results?

Yes

Is the language acceptable?

Yes

Do you have any ethical concerns with this paper?

No

Have you any concerns about statistical analyses in this paper?

No

Recommendation?

Accept with minor revision (please list in comments)

Comments to the Author(s)

The authors presented many figures which can be merge and decrease the numbers of figures. I made some corrections but I found the lack of writing flow so suggest to improve English writing and re-phrase most of the sentences. Although the result is excellent but due to the lack of writing flow it is hard for readers to get into it.

I suggest the paper for major revision with English re-editing (see Appendix A).

Decision letter (RSOS-191753.R0)

18-Feb-2020

Dear Dr Yong,

The editors assigned to your paper ("River capture in the middle reaches of the palaeo-Yarlung Zangbo River") have now received comments from reviewers. We would like you to revise your paper in accordance with the referee and Associate Editor suggestions which can be found below (not including confidential reports to the Editor). Please note this decision does not guarantee eventual acceptance.

Please submit a copy of your revised paper before 12-Mar-2020. Please note that the revision deadline will expire at 00.00am on this date. If we do not hear from you within this time then it will be assumed that the paper has been withdrawn. In exceptional circumstances, extensions may be possible if agreed with the Editorial Office in advance. We do not allow multiple rounds of revision so we urge you to make every effort to fully address all of the comments at this stage. If deemed necessary by the Editors, your manuscript will be sent back to one or more of the

original reviewers for assessment. If the original reviewers are not available, we may invite new reviewers.

- Data accessibility

If you wish to submit your supporting data or code to Dryad (<http://datadryad.org/>), or modify your current submission to dryad, please use the following link:
<http://datadryad.org/submit?journalID=RSOS&manu=RSOS-191753>

- Competing interests

- Authors' contributions

AB carried out the molecular lab work, participated in data analysis, carried out sequence alignments, participated in the design of the study and drafted the manuscript; CD carried out

the statistical analyses; EF collected field data; GH conceived of the study, designed the study, coordinated the study and helped draft the manuscript. All authors gave final approval for publication.

- Acknowledgements

- Funding statement

on behalf of Dr Mark Smith (Associate Editor) and Jon Blundy (Subject Editor)
openscience@royalsociety.org

Associate Editor's comments (Dr Mark Smith):

First comments:

An interesting paper that combines DEM analyses, geomorphological observations and Quaternary reconstruction to analyse river capture in the Himalayas.
I recommend this for review.

Second comments:

Thank you for submitting "River capture in the middle reaches of the palaeo-Yarlung Zangbo River" to Royal Society Open Science. I have received two reviews of your manuscript, which are included below and/or attached. Both reviews are relatively positive about the article; however, additional work is required to strengthen the fieldwork-derived evidence in some parts of the manuscript. Furthermore, both reviewers indicate that the geological setting of the study area needs to be communicated with greater clarity and that further English-language edits are made to the paper.

I am therefore returning the paper to you so that you can make the necessary changes, which fall under the category of 'Major Revisions'.

Comments to Author:

Reviewers' Comments to Author:

Reviewer: 1

Comments to the Author(s)

This paper combines fieldwork and DEM analysis to understand the development of the understudied middle reaches of the Yarlung Zangbo River; it combines geomorphology and sedimentology.

Currently, the evidence for the analysis is missing for some sections, especially the branch deposits. A new section in the results section is required to show the fieldwork that you

undertook. The implications of your work need to be included, how does this fit with the research further downstream?

Further analysis using the DEM could be used to strengthen your analysis, such as morphometric analysis of the river systems. Normalised steepness index and hypsometry can be used to show anomalous reaches of the river systems and can be completed using GIS. It would be interesting to see if there are any relationships between the branch river types and the main trunk river.

Specific comments below:

Page 1

Line 25 – Please expand on what you mean by Quaternary investigation – do you mean your field work? Or analysing the quaternary evolution of the catchment?

Line 28 – ‘it is believed’ – is this from previous published work? if so reword to ‘previous researchers have shown’

Line 34 – Water storage capacity is not the correct term, please look at the comments for page 6 line 11.

Page 2

Section 3 – This section currently has a tectonic / structural background of the study area, geological information on the geology (bedrock and superficial including the stratigraphic successions of the palaeolake infill). Include a geological map / stratigraphic chart earlier than Figure 7. Geology is referred to throughout the paper, so setting the context here would be useful.

This section could also benefit from information on the types of river channels you are looking at – are they bedrock or bedrock-alluvial or alluvial in nature? What is the current catchment area of the river system?

Line 51 – include the elevation decrease from east to west

Page 3

Line 57 – add what filling threshold you used

Page 4

Line 5 – what area / con value did you use?

Section 5 – you refer to the deposits of the branch outlets in the analysis section (Section 6.1) – however do not present the data in the results sections from your fieldwork. What are the character of the deposits? Did you see any clast imbrication that supports flow reversal. A new sub-heading in the results section is needed to display your data to support your later interpretations.

Page 4 /5

Section 5.2 – when each lake is introduced refer to the figure number (Figure 5), it would be useful to have a schematic diagram of the ages of the lakes and thickness of deposits – do you have a sediment logs of the area?

Page 6

Line 11 – Water storage capacity is not the right term here, sedimentary thicknesses can be high in areas of subsidence and low water volumes. Total accommodation space would be a more appropriate term to use.

Line 22 / 23 – ‘Melting water from alpine glaciers...’ – I would remove this sentence, it doesn’t add anything.

Line 25 – 41 – are there any age constraints on these surfaces via dating e.g., cosmogenic or apatite fission track?

Line 25 – 41 – the focus is on the c-c’ cross sections, add in information about all the cross sections extracted (a-a’ to e-e’)

Line 43 – 58 – references on bedrock channels are missing from this section.

For example, but not limited to:

Tinkler, K.J. and Wohl, E., 1998. Rivers over rock: Fluvial processes in bedrock channels (No. 107). American Geophysical Union. And references therein

Richardson, K. and Carling, P., 2005. A typology of sculpted forms in open bedrock channels (Vol. 392). Geological Society of America.

Line 44 – Reword ‘large drop’ to ‘steep gradient’ and include the gradient of the reach

Line 48 – Reword ‘There are no quaternary deposits in the gorge due to the strong erosional downcutting of the river...’ In a bedrock setting, deposits are not expected in the reach due to the high stream powers causing high transport capacity of the water.

Page 7

Section 6 in general– what are the implications of your study? Do any of the dates correspond with information in the depositional basin? Does this support previously published research on the lower reaches or not?

Section 6.1 – the evidence is needed in the results section to support your interpretations for this section.

Section 6.2 – how does your results and interpretation fit with the other work further downstream cited in the introduction?

Page 8

Line 5 – do you have a reference for the elevation of the plateau?

Line 17 – large water capacity, see comment for page 6 line 11

Page 9

Line 14 – ‘were burst by the river’s upstream development’ reword to ‘drained by headward erosion’

Line 31 – water storage capacities, see comment for page 6, line 11.

Figures

Figure 1 – The text in the inset box should be increased, when printed, this is hard to read, a legend is needed for the DEM in the inset box.

What do the yellow dashed line refer to? Missing from the key

Figure 2 – it would be useful to include information on the thresholds you used

Figure 3 – Move the inset b, below part A – it is currently too small to see and could be made much larger if you place it below A

Figure 4 – place a white box behind the scale, and change the text to black / bold the legend – it is currently hard to see

Figure 5 – you could add the sediment logs in this figure for B, C and D (See comment for section 5.2)

Figure 7 – the red box in inset A is hard to see with the current colour scheme used

Figure 8 – are the moraines mentioned in section 5.3 visible in any of the cross sections?

Figure 14 – in the text you mention about draining of the lakes by headward erosion / shrinkage of the lakes, it would be good to show this in the schematic as section C

Figure 15 – reword caption to ‘Satellite image of the Qusong Drainage Basin’ – or add a scale for the elevation

Reviewer: 2

Comments to the Author(s)

The authors presented many figures which can be merge and decrease the numbers of figures. I made some corrections but I found the lack of writing flow so suggest to improve English writing

and re-phrase most of the sentences. Although the result is excellent but due to the lack of writing flow it is hard for readers to get into it.

I suggest the paper for major revision with English re-editing.

Author's Response to Decision Letter for (RSOS-191753.R0)

See Appendix B.

RSOS-191753.R1 (Revision)

Review form: Reviewer 1

Is the manuscript scientifically sound in its present form?

No

Are the interpretations and conclusions justified by the results?

No

Is the language acceptable?

Yes

Do you have any ethical concerns with this paper?

No

Have you any concerns about statistical analyses in this paper?

No

Recommendation?

Major revision is needed (please make suggestions in comments)

Comments to the Author(s)

The manuscript has improved and the figures are easier to see, the addition of the new section 5.2 is welcomed. However, in its current form it is not sufficient to back up your analysis. There is no actual data presented in your paper - sedimentary logs, photos of the deposit, imbrication data. In the previous version 'Figure 5' could have been expanded to add this - which has now been removed. Without the actual evidence, it is hard to accept the descriptions now added in Section 5.2.

The interplay between climate and tectonics is also really interesting, and could be expanded on to show wider global implications.

Specific comments below:

Section 5.2 - Geology information is needed to set the context, currently this just looks at the geomorphic setting of river types and the tectonic setting. Figure 4 includes information on geology (rock types) -add some text about the geology here.

Page 9, line 7 - reword 'the river bed is composed of bedrock due to strong erosion' to 'the bedrock gorges are areas of high erosion rates and stream power'

Page 9, line 9 - reword largest to deepest gorge

Page 11, line 10 - change 'extrusion' to 'convergence'

Page 11, line 25 - just put '87%', the decimal place isn't needed

Page 12, line 39 to 41 - 'this finding further indicates....' - this is interpretation and shouldn't be in the results section

Section 5.2 - More evidence is needed in this section of your fieldwork - where is the evidence / data of imbrication? Did you take a picture? What was the strike/dip? how many clasts did you measure, can you create a stereonet?

A sedimentary log is needed with pictures of the different facies types (e.g., sand lenses, imbrications)

Do you have other clast data - lithology?

Page 14, line 34 - it would be good to bring the imbrication in here

Page 15, line 31 - reword largest to deepest

Page 15, line 37 - reword continental river to 'endorheic basin'

Page 16 - there is an interesting interplay between tectonics and climate influencing the basin, a paragraph should be added to highlight this - this will show wider implications of the study

Conclusions - It would be good to say how this relates to the other research and bring back the information cited in the introduction. Even if the sentence is to say, how your work has added to the story 'this research has confirmed the work downstream (REFS)' for clarity.

Figure 1 - the DEM legend needs units (m)

Figure 2 - you could increase the size of the D and E insets to make them bigger and use all of the white space

Figure 7 - the white text is still hard to read, either have a box behind the text (white box, black text) or change the text to a different colour and bold it

Figure 8 - pop a white box behind the scale

Figure 10 - Can you add any rough time indications of the evolution? - annotate the stream capture on E. The 'Jiacha mountain' is not referred to in the text - it is currently hard to place C - E in the context of A and B.

Decision letter (RSOS-191753.R1)

31-Mar-2020

Dear Dr yong:

On behalf of the Editors, I am pleased to inform you that your Manuscript RSOS-191753.R1 entitled "River capture in the middle reaches of the palaeo-Yarlung Zangbo River" has been accepted for publication in Royal Society Open Science subject to minor revision in accordance with the referee suggestions. Please find the referees' comments at the end of this email.

The reviewers and Subject Editor have recommended publication, but also suggest some minor revisions to your manuscript. Therefore, I invite you to respond to the comments and revise your manuscript. Please ensure that you fully address these remaining concerns, as publication is contingent on the Editors being satisfied by the changes made.

- Ethics statement

- Data accessibility

It is a condition of publication that all supporting data are made available either as supplementary information or preferably in a suitable permanent repository. The data

accessibility section should state where the article's supporting data can be accessed. This section should also include details, where possible of where to access other relevant research materials such as statistical tools, protocols, software etc can be accessed. If the data has been deposited in an external repository this section should list the database, accession number and link to the DOI for all data from the article that has been made publicly available. Data sets that have been deposited in an external repository and have a DOI should also be appropriately cited in the manuscript and included in the reference list.

If you wish to submit your supporting data or code to Dryad (<http://datadryad.org/>), or modify your current submission to dryad, please use the following link:
<http://datadryad.org/submit?journalID=RSOS&manu=RSOS-191753.R1>

- **Competing interests**

- **Authors' contributions**

- **Acknowledgements**

- **Funding statement**

Because the schedule for publication is very tight, it is a condition of publication that you submit the revised version of your manuscript before 09-Apr-2020. Please note that the revision deadline will expire at 00.00am on this date. If you do not think you will be able to meet this date please let me know immediately.

on behalf of Dr Mark Smith (Associate Editor)
openscience@royalsociety.org

Associate Editor Comments to Author (Dr Mark Smith):

Many thanks for addressing the reviewers' comments in this revised version of the manuscript. You will note that a reviewer still has legitimate concerns regarding the evidence relating to the sedimentary deposits. Moreover, the reviewer has provided a short list of very specific revisions.

Subject to minor revisions (notably an expanded section 5.2 detailing sedimentary evidence but also the specific points) this material should be suitable for publication in RSOS.

Best wishes,
Mark

Associate Editor: 2

Comments to the Author:

Many thanks for these revisions. Given the nature of the changes (including a new section) I would like to consult the reviewer who suggested the changes.

Reviewer comments to Author:

Reviewer: 1

Comments to the Author(s)

The manuscript has improved and the figures are easier to see, the addition of the new section 5.2 is welcomed. However, in its current form it is not sufficient to back up your analysis. There is no actual data presented in your paper - sedimentary logs, photos of the deposit, imbrication data. In the previous version 'Figure 5' could have been expanded to add this - which has now been removed. Without the actual evidence, it is hard to accept the descriptions now added in Section 5.2.

The interplay between climate and tectonics is also really interesting, and could be expanded on to show wider global implications.

Specific comments below:

Section 5.2 - Geology information is needed to set the context, currently this just looks at the geomorphic setting of river types and the tectonic setting. Figure 4 includes information on geology (rock types) -add some text about the geology here.

Page 9, line 7 - reword 'the river bed is composed of bedrock due to strong erosion' to 'the bedrock gorges are areas of high erosion rates and stream power'

Page 9, line 9 - reword largest to deepest gorge

Page 11, line 10 - change 'extrusion' to 'convergence'

Page 11, line 25 - just put '87%', the decimal place isn't needed

Page 12, line 39 to 41 - 'this finding further indicates....' - this is interpretation and shouldn't be in the results section

Section 5.2 - More evidence is needed in this section of your fieldwork - where is the evidence / data of imbrication? Did you take a picture? What was the strike/dip? how many clasts did you measure, can you create a stereonet?

A sedimentary log is needed with pictures of the different facies types (e.g., sand lenses, imbrications)

Do you have other clast data - lithology?

Page 14, line 34 - it would be good to bring the imbrication in here

Page 15, line 31 - reword largest to deepest

Page 15, line 37 - reword continental river to 'endorheic basin'

Page 16 - there is an interesting interplay between tectonics and climate influencing the basin, a paragraph should be added to highlight this - this will show wider implications of the study

Conclusions - It would be good to say how this relates to the other research and bring back the information cited in the introduction. Even if the sentence is to say, how your work has added to the story 'this research has confirmed the work downstream (REFS)' for clarity.

Figure 1 - the DEM legend needs units (m)

Figure 2 - you could increase the size of the D and E insets to make them bigger and use all of the white space

Figure 7 - the white text is still hard to read, either have a box behind the text (white box, black text) or change the text to a different colour and bold it

Figure 8 - pop a white box behind the scale

Figure 10 - Can you add any rough time indications of the evolution? - annotate the stream capture on E. The 'Jiacha mountain' is not referred to in the text - it is currently hard to place C - E in the context of A and B.

Author's Response to Decision Letter for (RSOS-191753.R1)

See Appendix C.

Decision letter (RSOS-191753.R2)

03-Apr-2020

Dear Dr yong,

It is a pleasure to accept your manuscript entitled "River capture in the middle reaches of the palaeo-Yarlung Zangbo River" in its current form for publication in Royal Society Open Science.

on behalf of Dr Mark Smith (Associate Editor)
openscience@royalsociety.org

Appendix A**ROYAL SOCIETY
OPEN SCIENCE****River capture in the middle reaches of the palaeo-Yarlung
Zangbo River**

Journal:	Royal Society Open Science
Manuscript ID	RSOS-191753
Article Type:	Research
Date Submitted by the Author:	18-Oct-2019
Complete List of Authors:	yong, liu; Chengdu University of Technology, Wang, Yunsheng; Chengdu University of Technology Wei, Liangshuai; CAGS Institute of Exploration Technology Shen, Tong; Chengdu University of Technology Su, Qinfeng; CAGS Institute of Exploration Technology Huang, Anbang; CAGS Institute of Exploration Technology Jia, Yi; CAGS Institute of Exploration Technology
Subject:	Geology < EARTH SCIENCES
Keywords:	DEM, plate collision, palaeolakes, Jiacha Gorge, watershed, flow reverse
Subject Category:	Earth science

Author-supplied statements

Relevant information will appear here if provided.

Ethics

Does your article include research that required ethical approval or permits?:

This article does not present research with ethical considerations

Statement (if applicable):

CUST_IF_YES_ETHICS :No data available.

Data

It is a condition of publication that data, code and materials supporting your paper are made publicly available. Does your paper present new data?:

Yes

Statement (if applicable):

The graphical data of Planation surface and valley shoulder elevation in different parts of the middle reaches of YZR and the river network flow directions that supporting this paper have been uploaded as electronic supplementary material.

Conflict of interest

I/We declare we have no competing interests

Statement (if applicable):

CUST_STATE_CONFLICT :No data available.

Authors' contributions

This paper has multiple authors and our individual contributions were as below

Statement (if applicable):

Yong Liu, Liangshuai Wei, Tong Shen, Qinfeng Su, Anbang Huang ang Yi Jia conducted the experiment and data analysis, Yunsheng Wang guided the writing process and Yong Liu wrote the paper.

River capture in the middle reaches of the palaeo-Yarlung Zangbo River

Yong Liu^{1,2*}, Yunsheng Wang¹, Liangshuai Wei², Tong Shen¹, Qinfeng Su²,
Anbang Huang², Yi Jia²

1. State Key Lab. of Geo-Hazard Prevention and Geo-Environment Protection, Chengdu University of Technology, Chengdu 610059, China

2. Institute of Exploration Technology, Chinese Academy of Geological Sciences, Chengdu 710043, China

Keywords: DEM; plate collision; palaeolakes; Jiacha Gorge; watershed; flow reverse.

1. Abstract

There are 51 branches in the middle reaches of the Yarlung Zangbo River (YZR), and 87% of the branches west of Jiacha Gorge are inverse cross rivers and orthogonal rivers, reflecting an anomalous development of branches. In this paper, Quaternary investigation and digital elevation model (DEM) methods were used to analyse the causes of this anomalous phenomenon. It is believed that the initial elevation of the Qinghai-Tibet Plateau, formed by collision between the Indian plate and the Eurasian plate, was not high and that precipitation was abundant. To the west of Jiacha Gorge, the palaeo-YZR flowed westward before the early Pleistocene and merged into Zada, Zhongba, Jilong and Gamba-Dingri palaeolakes with large water storage capacities on the west side to form a continental river. The palaeo-YZR east of Jiacha Gorge (RE) flowed eastward, which is consistent with the current river flow. Additionally, there was a watershed in the area of Jiacha Gorge. With the intensification of the collision between the Indian plate and the Eurasian plate, the Qinghai-Tibet Plateau rapidly uplifted and formed a high western and low eastern terrain, promoting erosional downcutting of the RE and its westward development. During the early Pleistocene, RE crossed the watershed and captured the palaeo-YZR west of Jiacha Gorge (RW), causing the reversal of the flow direction of the RW.

2. Introduction

Yarlung Zangbo River (YZR) is the highest river in the world and has developed along the collision zone between the Indian plate and Eurasian plate. In recent years, the river capture of the YZR Basin has attracted the attention of many scholars. A large number of research results show that YZR was formed after the early convergence of rivers between the Garo Mountains and the Himalayas under the action of crustal uplift (Wang et al., 2002; Gansser, 1964; Pan et al., 1990; Parrish et al., 1993; Coleman et al., 1995; Harris, 1995; Tapponnier, 1981). Wang et al. (2002) studied the climate and topography of the lower reaches of the YZR and concluded that the Brahmaputra River located in the tropical

* Author for correspondence (1039786137@qq.com).

State Key Lab. of Geo-Hazard Prevention and Geo-Environment Protection, Chengdu University of Technology, Chengdu 610059, China

monsoon belt of the Indian Ocean traced to the north and captured the YZR, resulting in a sudden southward mutation of the YZR downstream. Chen et al. (2008) considered the Yarlung Zangbo Grand Gorge and its upper reaches to belong to different river systems; at approximately 30 ka, the Yarlung Zangbo Grand Gorge was a branch of the Palong Zangbo River, which traced upstream and captured the palaeo-YZR. Laura et al. (2015) established a structural model and found that the early Miocene sediments in Sulma Basin of the Indian plate originated mainly from the Asian plate through the rivers in the eastern Himalayas traced to the north that captured the YZR. Vance et al. (2011) analysed the digital elevation models (DEMs) from ASTER (30 m) and the Shuttle Radar Topography Mission (SRTM) (90 m) and found that there were some signs of river capture in the lower reaches of the Himalayas. Shi et al. (2010) believed that the nearly east-west trending palaeo-YZR developed in the Pliocene captured the nearly north-south trending rivers developed in the Eocene-Miocene, forming the present near-direct water system, and these north-south trending rivers became the branches of the YZR, as determined by remote sensing analysis, geological survey and chronological study. Burrard and Hayden found that the main branches of the YZR flowed in the opposite direction into the main stream in 1907, which suggested that the YZR flowed from east to west before it was connected, contrary to the current flow direction. Li (1954) considered that the rising speed of the plateau surface was different according to the Quaternary history throughout the country, which results in the capture of rivers, and the reversal of the YZR flow direction was more or less affected by the crustal rise. Yang (1982) considered that the YZR, including the Grand Gorge, was a precursor river adapted to different structural faults based on field investigation data and geological structure analysis; the Grand Gorge was not caused by river capture, and the YZR did not flow from east to west in the past.

At present, these and other studies emphasize river capture in the lower reaches of YZR, while river capture in the middle reaches of YZR is rarely reported. There is a lack of systematic and comprehensive study on river capture in the middle reaches of YZR, and it is of great significance for the study of the tectonic evolution process in the YZR Basin. In this paper, the process of river capture and its evolution in the middle reaches of the palaeo-YZR were analysed by field geomorphology, Quaternary investigation, DEM and comprehensive data analysis.

3. Study area and geological background

The YZR originates from Majieyangzom glacier and has an average elevation higher than 4000 m, making YZR the highest river in the world. The river is located along the east-west fault zone between the Himalaya Mountains and the Gangdise Mountains and flows southward around Namjag Barwa Peak. Wide valleys and gorges are alternately distributed in the basin. The elevation of the riverbed decreases gradually from west to east. The middle reaches of the YZR extend from Zhongba County in the west (30° 13' N, 83° 14' E) to the town of Pai in the east (29° 30' N, 94° 52' E) with a distance of approximately 1184 km (Fig. 1).

The collision zone between the Eurasian plate and the Indian plate is known as the Yarlung Zangbo Suture Zone. The collision between the Eurasian plate and Indian plate mainly underwent the initial collision stage in the early Palaeocene (approximately 65 Ma),

the comprehensive collision stage at the end of the Palaeocene (approximately 55 Ma) and the
continuous collision stage at the end of the Eocene (40-38 Ma). Initial collision and
comprehensive collision mainly occurred in the protruding part of the Indian plate front. The
two plates were completely closed during the continuous collision stage, and the Tethys
Ocean between the two plates disappeared at this time. The collision of the three stages
caused the crust to rise to form the initial Qinghai-Tibet Plateau (Pan, 1999). In the late Eocene,
the initial Qinghai-Tibet Plateau began to converge strongly inland, causing the plateau to
rise rapidly and the relative movements such as subduction, napping, slipping and stretching
to appear at this time (Liu et al., 2018).

Collision between the Indian plate and the Eurasian plate first occurred in the middle of
the collision zone and then in the east and west (Ding et al., 2009). With the continuous rise of
the middle of the collision zone, a north-south uplift zone with a width of approximately 120
18 km was formed, and some piedmont plains and molasses were developed on both sides
of the uplift zone (Ge et al., 2006).

4. Methods

4.1 DEMs and pre-processing of the river network

The DEMs used in this study were derived from the SRTM data. SRTM is a joint survey
conducted by NASA, the National Imagery and Mapping Agency (NIMA), and German and
Italian space agencies and was carried out by the SRTM system on the space shuttle
Endeavour launched by the United States. The SRTM data contain approximately 80% of the
global land surface from 60°S to 60°N and were published in late 2007 with C band and X
band radar (Farr et al., 2007). The band width of the X band is narrower, resulting in a smaller
coverage. Therefore, the SRTM data we used in this study belong to the C band. There are
three versions of the SRTM elevation data: SRTM1, SRTM3 and SRTM30. SRTM1 with a
resolution of 1-arc-second, i.e., 30 m, only covers the United States; SRTM3 with a resolution
of 3-arc-seconds, i.e., 90 m, covers the whole world; and SRTM30 with a resolution of
30-arc-seconds, i.e., 1000 m, also covers the whole world. At present, the SRTM3 data are
available in China. The digital elevation was generated from the SRTM data using
interferometric synthetic aperture radar (InSAR). The elevation base of SRTM3 is the geodetic
plane of EGM96, and the plane base is WGS84. The nominal absolute elevation accuracy is \pm
16 m, and the absolute plane accuracy is \pm 20 m (Tachikawa et al., 2011). All digital elevation
data for this study were collected from the China Geospatial Data Cloud (CGDC). Further
details of the SRTM data are listed in Table 1.

ArcGIS 10.2 software was used to read the acquired digital elevation data and generate
the DEM. The DEM was pre-processed using the hydrological calculation module in the
software to extract the river network in the middle reaches of the YZR, and the specific
process is shown in Figure 2. First, the AGREE method is used to smooth the DEM so that the
processed DEM is consistent with the input vector elevation data; then, the DEM depressions
were filled by calculating the depth and contribution rate of the depressions and setting a
reasonable filling threshold in conjunction with the real geomorphological conditions.
Depressions less than the threshold were filled, and depressions greater than the value were
retained. The D8 method is used to calculate the flow direction of the river, that is, to

calculate the steepest direction between each grid and its adjacent grid. Then, the catchment
area of the upper reaches of the grid is calculated to extract the river network, which is shown
in Figure 3 A. Based on the elevation of the DEM, a river profile of the middle reaches of YZR
is shown in Figure 3 B.

4.2 Data collection

Stratigraphic data in the middle reaches of the YZR are collected from a 1:50,000 regional
geological survey, for which the stratigraphic dating was completed at the State Key
Laboratory of Continental Dynamics, Northwest University, China. After the samples taken
from the stratigraphic profiles were pulverized in the laboratory, they were separated by
flotation and electromagnetic methods. Finally, zircons with a relatively complete shape, no
cracks, and no inclusions were selected for measurement under a binocular lens. The selected
zircon samples were fixed in a colourless transparent epoxy resin and polished to expose a
plane, which can be used to determine the age of formation using the LA-ICP-MS U-Pb
method.

5. Results

5.1 Characteristics of the river network

The river network map of the middle reaches of the YZR (Fig. 3 A) shows that the YZR
and its tributaries are substantially deformed by plate extrusion, and the whole river network
is narrow in the south-north direction and wide in the east-west direction. The directions of
the primary branches flowing into the YZR are not the same. For example, the Nianchu River
and Lhasa River flow into the YZR in the inverse direction, while the Niyang River flows into
the YZR in the consequent direction (Fig. 4). The directions of the branches flowing into the
YZR are shown in Table 2. There are a total of 31 branches located west of Jiacha Gorge. On
the left bank, 14 branches flow into the YZR in the reverse direction, 5 branches flow into the
YZR in the orthogonal direction, and the other 4 branches flow into the YZR in the
consequent direction. All 8 branches on the right bank are inverse cross rivers. To the east of
Jiacha Gorge, there are a total of 20 branches. Among them, 6 branches on the left bank flow
into the YZR in the consequent direction, and the other 2 branches flow into the YZR in the
inverse direction. The 11 branches on the right bank flow into the YZR in the consequent
direction, and the other 1 branch flows into the YZR in the inverse direction. A total of 87.1%
of branches west of Jiacha Gorge are inverse cross rivers and orthogonal rivers, and 85% of
the branches east of Jiacha Gorge are consequent rivers.

From the river profile (Fig. 3 B), the elevation of the riverbed gradually decreases from
the west at 4500 m to the east at 1020 m. From Zhongba County to Jiacha Gorge, the average
longitudinal gradient of the riverbed is 0.7‰, and the average longitudinal gradient of the
riverbed from Jiacha Gorge to the town of Pai is 2.5‰. The elevation of the riverbed in
Yarlung Zangbo Grand Gorge drops sharply, and the longitudinal gradient of the riverbed
can reach up to 10.3‰.

5.2 Palaeolakes on the western Qinghai-Tibet Plateau

In recent years, it has been found that there are several palaeolakes located on the

western Qinghai-Tibet Plateau, such as Zhada, Zhongba, Jilong, Dati and Gangba-Dingri Palaeolakes (Fig. 5). Among them, Zhada and Zhongba Palaeolakes are located west of the YZR suture zone, Jilong, Dati, and Gangba-Dingri Palaeolakes are located southwest of the YZR suture zone and north of the Himalayas. These palaeolakes were formed at 7.0-1.36 Ma and have undergone the evolutionary process of formation-expansion-shrinkage-disappearance (Liu et al., 2007; Han et al., 2017; Wang et al., 1996; Zhu et al., 2006; Wang et al., 2018; Deng et al., 2015; Du et al., 2015).

Zhada Palaeolake (30°50'-32°20' N, 79°00'-80°30' E) is located along the NW-SE direction and is strip-shaped, with a width of approximately 70 km and a length of approximately 260 km. The lake basin basement is composed of pre-Jurassic sandstone and limestone and is covered by fluvial-lacustrine sediments with a thickness of nearly 1,000 m; the area of the sediments is as large as 9000 km² (Han et al., 2011; Wang et al., 2018). These characteristics indicate that the water capacity of the palaeolake can be estimated to reach 9×10¹² m³. Saylor et al. (2016) found that the sedimentary profiles of Zhada Palaeolake had fluvial-supratidal, littoral, fluvial and delta sediments from bottom to top. This finding reflects a complete evolutionary process from formation and expansion to shrinkage (Rea, 1992; Brookfield, 1993). Zhu et al. (2006) measured the age of the palaeolake sediments and found that Zhada Palaeolake began to form at approximately 5.4 Ma and began to shrink at approximately 3.2 Ma.

Zhongba Palaeolake (29°21'-30°12' N, 82°18'-83°24' E) is located along the NW-SE direction, with a width of approximately 34 km and a length of approximately 126 km, and the palaeolake basin covers an area of 4284 km². Liu et al. (2007) acquired a lacustrine sediment profile of 43 m ~~deep in the basin~~. Through sampling and dating of the sediments from the profiles, the palaeolake sediments began to be deposited at **approximately 4 Ma**. During this period, the lacustrine delta sedimentary facies play an important role in the deposition process, and the sediments **were** mainly calcareous and sandy mudstone. At approximately 3.45 Ma, the lake surface reached its maximum area and began to shrink, and the salinity of the lake water and the content of Limnocytherellina in the sediments **increased**.

Jilong Palaeolake Basin is located on the South Bank of YZR (28°18'-29°17' N, 84°29'-86°05' E) and is narrow in the north and wide in the south, with an area of approximately 300 km². The sediments in the basin are approximately 300 m thick and consist of ~~the following from the bottom to the top~~: lacustrine delta sediments, littoral sediments, braided channel sediments, and littoral and alluvial sediments. These sediments **reflect the stages of** the formation, expansion, shrinkage and disappearance of Jilong Palaeolake. Palaeomagnetic dating of sedimentary profiles shows that the Jilong Palaeolake was formed at approximately 7 Ma, began to shrink after 3.4 Ma, and disappeared at 1.7 Ma (Wang et al., 1996).

The Gangba-Dingri Basin is located in the northern foot of the Himalayas and is adjacent to the YZR suture zone (28°12'-28°48' N, 87°05'-88°03' E). This basin is approximately 900 km from east to west and approximately 50-70 km wide from north to south, with an area of approximately 70,000 km². Du et al. (2015) found that a large number of thick shales were deposited in the basin by field investigation, **indicating that the area was once a lake environment**. The Gangba-Dongshan Formation ~~shale~~ formed in the Miocene is widely distributed, mostly in Gangba and Dingri county. This formation ~~is~~ composed of grey-black thin layered shale and silty shale. These shale layers contain fossils such as ammonite and

double-shell and are 190-1100 m thick. These characteristics indicate that the Gangba-Dingri Palaeolake had a deep water depth, a large distribution area and a large water capacity.

In summary, the palaeolakes distributed west of the Qinghai-Tibet Plateau reached the largest scale at 3.2-3.45 Ma and began to shrink, appearing at approximately 1.7 Ma. These palaeolake sediments have a wide distribution area and large sediment thickness, indicating that the water storage capacity was high.

5.3 Geomorphological characteristics of the Jiacha Gorge

In the field investigation, there are many moraine terraces located on the left bank of Woka River, which is distributed along the west side of Jiacha Gorge (Fig. 6). The surface of the terraces is relatively flat and inclines to the west. The thickness of the moraine terraces is 80-200 m. Eroded by the Woka River, many steep slopes at 45°-70° formed on the front edge of the terraces. The moraine terraces are caused by the westward movement of glaciers along the eastern side of Jiacha Gorge. The sediments of the moraine terraces are poorly sorted and have good water permeability. Melting water from alpine glaciers infiltrates the moraine sediments to form wetlands, which are rare on the Qinghai-Tibet Plateau. The Woka River and its adjacent Zengqi River originated from the glacier on the eastern side of Jiacha Gorge.

Cross-sectional maps of the YZR valley west of Jiacha Gorge (RW) are extracted, as shown in Figure 7 and Figure 8. There are many third-level planation surfaces and first-stage valley shoulders located in the RW. Among them, the elevation of the third-level planation surfaces is 3857-4337 m, and the average elevation of the planation surfaces on the north bank is higher than that on the south bank. From Jiacha Gorge to Sangri County (near the c-c' section), the elevation of the third-level planation surfaces gradually decreases. From the c-c' section to the west, the elevation of the third-level planation surfaces gradually increases. The elevation of the first-stage valley shoulders is 3670-3841 m, and the average elevation of the first-stage valley shoulders on the north bank is lower than that on the south bank. From east to west, the elevation trend of the first-stage valley shoulder is consistent with the trend of the third-level planation surfaces, exhibiting a "V"-shaped trend of decreasing first and then increasing. The elevation of the third-level planation surface and the first-stage valley shoulder is the lowest in Sangri County (Fig. 9).

Jiacha Gorge and its downstream reaches are dominated by meandering rivers. The river channel of the Jiacha Gorge section is characterized by a large drop, a fast flow speed and a strong ability to transport river material. The river water strongly erodes and cuts the riverbed, causing the river valley to deepen and form a narrow and deep "V" shaped valley. Because of the strong erosional downcutting of the river water, there are almost no Quaternary sediments located in Jiacha Gorge, and many rock sills, potholes and deep troughs have formed in the riverbed (Fig. 7 B). The two sides of the river valley are Cretaceous granite, with a hard texture and a strong resistance to weathering and erosion, and the slopes of the two sides are very steep, with a slope greater than 50°. Zhu et al. (2011) found a large number of terraces in the downstream reaches of Jiacha Gorge, and the maximum terrace series reached 9 grades. From the phase map of river terraces along the Jiacha sector of the YZR (Fig. 10), the elevation of the terraces gradually decreases from west to east.

6. Discussion

6.1 Analysis of anomalous deposits at the branch outlets

The sediments transported by the branches will be affected by the YZR during their deposition process at the branch outlets and will exhibit the hydrodynamic characteristics of the YZR. However, the hydrodynamic characteristics of the sediments at the outlets of Menqu River and Nanchu River are inconsistent with those of the YZR.

Menqu River is an inverse cross branch of the YZR, flowing into the YZR from southeast to northwest (Fig. 11 A). In the field investigation, the fluvial sediments transported by the branch are mainly deposited on the west side of the outlet, and the thickness of the fluvial sediments ~~can reach as large as 127.8 m~~ (Fig. 11 B and Fig. 11 C). Based on hydrodynamic characteristics, the **branch sediments** will be transported by the YZR. **When the YZR flows eastward, the branch sediments will be transported to the east side of the branch outlet for deposition (Fig. 12 A); when the YZR flows westward, the branch sediments will be transported west for deposition (Fig. 12 B).** These findings indicate that the YZR flowed westward in the past and lasted for a long time. After the YZR turned eastward, some sediments deposited on the west side of the branch outlet were eroded, and a large amount of sediments have been preserved to date.

Nianchu River is also an inverse cross branch, flowing into the YZR from southeast to northwest (Fig. 13 A). ~~In the field investigation, the sediments on the west side of the Nianchu River outlet are much higher than those on the east side, and a steep ridge with a height of approximately 2.4 m appears at a distance of 1-1.5 km from the west side. This finding indicates that the sediments transported by the branch have undergone a dramatic change during their deposition process, which should be affected by the YZR (Fig. 13 B). When the YZR flowed westward, the branch sediments deposited on the west side of the outlet formed a higher sedimentary platform under the long-term hydrodynamic forces. After the YZR turned eastward, the eastern bank of the outlet was eroded by the YZR and the branch sediments deposited on the eastern side formed a new broad and gentle sedimentary platform.~~

6.2 Analysis of river network formation mechanism

YZR formed **after the collision between** the Indian plate and the Eurasian plate and has been active since the Cenozoic. The Indian plate **subducted northward** at approximately 50 Ma, causing the ~~Gangdise~~ Gangdise Mountains to rise. Meanwhile, many rivers originating from the Gangdise Mountains and the northern Tibetan Plateau flowed southward to the Indian continent due to **the topography** (Fig. 14 A). ~~As the Indian plate continued to subduct northward, the Himalayas began to uplift and block the rivers flowing south, and the rivers originating from north of the Himalayas and south of the Gangdise Mountains began to converge between the two mountains.~~ During **this** period, **faults and folds occurred** in the marine sediments formed in the Palaeocene and the early-middle Eocene and rose with the Himalayas, resulting in the formation of several large E-W-trending troughs and valleys in the YZR suture zone. On the landscape, these troughs and valleys are undulating and beaded, receiving sediments and river water from both sides (Qayyum et al., 1997; Xiang et al., 2002; Ding et al., 1999). **The river water from both sides converged in the troughs and valleys to form lakes.** The **rapid uplift** of the Qinghai-Tibet Plateau began at the end of the late Tertiary

period of ~~3.4 Ma~~. Before this period, the elevation of the Qinghai-Tibet Plateau was below
2000 m, and there was a large north-south uplift zone located in the middle of the suture zone.
Influenced by the Indian Ocean current, the rainfall was abundant, and the rivers between the
two mountains have high flow rates and strong erosional forces (Zhu et al., 2011; Zhang et al.,
2008). After the ridges between the troughs and valleys were eroded by water flow,
the lakes were connected to form the early prototype of the YZR (Fig. 14 B) (DAVID, 1998; Han et al.,
2017). Affected by the uplift of the middle of the suture zone, the elevation of Jiacha
Mountain was higher than that of the two sides, causing the rivers west of Jiacha Mountain to
flow west, while the rivers east of Jiacha Mountain flowed east. Since there was no outflow
river and its fluvial sediments were deposited on the west side of the suture zone, the river
flowing westward should be a continental river. The palaeolakes with large water capacities
on the west side should be the final destination for the rivers flowing westward. While
flowing westward, the RW not only captured the rivers flowing north and south but also
produced a number of new branches flowing into the YZR. Therefore, most branches west of
Jiacha Gorge are inverse cross rivers and orthogonal rivers. The abnormal relationship
between the branches and the RW could not have been caused by the fault movement; if the
left bank branches flowed west due to fault movement, the right bank branches would have
flowed east due to the fault movement opposite to that on the left bank. However, most of the
branches on both banks flowed west or perpendicular to the YZR, and faults that would have
influenced the flow direction of the branches were not found in the field investigation.

6.3 Formation time of Jiacha Gorge

Jiacha Mountain is located in the tectonic uplift zone and is covered by glaciers. Before
the formation of Jiacha Gorge, there was a watershed separating the water systems on the
east and west sides. Therefore, the elevation of the planation surface, valley shoulder and
terrace formed by the rivers on both sides of the watershed gradually decreased from the
middle to both sides. At approximately 3.4 Ma, as the intra-continental convergence caused
by the collision between the India plate and the Eurasian plate intensified, the Himalayas
continued to rise rapidly, blocking the Indian Ocean current from flowing north, resulting in
a decrease in precipitation in the YZR Basin. Meanwhile, the high western and low eastern
terrains were formed by the differential uplift of the Himalayas (Qayyum, et al., 1997; Xiang,
2002). Therefore, during this period, the palaeolakes located west of the YZR began to shrink,
and the planation surface, valley shoulder and terrace of the YZR also began to rise. As the
west side of the YZR Basin rose faster, the elevation of the planation surface and valley
shoulder west of Jiacha Gorge exhibited a "V"-shaped trend from west to east, which first
decreased and then rose.

Uplift of the Qinghai-Tibet Plateau caused the erosional downcutting of the RE to
gradually increase. After the RE crossed Jiacha Gorge, it captured the RW, causing the RW to
flow eastward. Therefore, the palaeolakes on the west side of the YZR could not have been
supplied by the YZR, resulting in the disappearance of the palaeolakes. According to the
dating results of the lake sediments, the palaeolakes located on the west side of the YZR
disappeared at approximately 1.36 Ma. That is, Jiacha Gorge was formed at approximately
1.36 Ma. In the field investigation, two small palaeolakes were also found in the mountain
basin on the west side of Jiacha Gorge, which were located in the upper reaches of the

Qusong River (Fig. 7 C and Fig. 15). According to a typical lake sediment profile located in
Qusong River completed by a 1:250000 regional geological survey (Fig. 16), the lake
sediments were formed in the early Pleistocene (Q1), and the formation time was determined
by the **sporopollen** in the lake sediments. From the topography, the area where the two
palaeolakes were located was the river source area of the RW before Jiacha Gorge was formed,
which had a low river flow rate and weak erosional forces. After Jiacha Gorge formed, the
RW was captured by the RE, resulting in an increase in the flow rate of the rivers in the Jiacha
area. Under this effect, erosion by Qusong River increased sharply, and the palaeolakes in the
upper reaches were burst by the river's upstream development. This process took place in Q1,
indicating that Jiacha Gorge was formed and the RW was captured by the RE in Q1.

17 7. Conclusions

Most branches of the middle reaches of the YZR are inverse cross rivers and orthogonal
rivers, reflecting an anomalous development of the branches. Based on Quaternary
investigation and DEM methods, this paper analyses the area and draws the following
conclusions:

(1) Collision between the Indian plate and the **Eurasian plate first occurred in the middle**
**of the collision zone and then in the east and west**, resulting in the formation of a north-south
uplift zone in the area of Jiacha Mountain in the middle of the collision zone, which became a
natural watershed that separated the river network on the east and west sides.

(2) Before Jiacha Gorge ~~was formed~~, the RW flowed westward into the palaeolakes with
large water storage capacities, ~~such as Zada, Zhongba, and Jilong palaeolakes~~ on the west
side of the YZR; the RE flowed eastward. During the westward flow, the RW captured the
rivers from north and south in the YZR valley and generated some new branches, resulting in
many inverse cross rivers and orthogonal rivers in the middle reaches of the YZR.

(3) Jiacha Gorge was formed in the early Pleistocene, causing the RE to capture the RW
and the RW to flow eastward. Due to the westward flow of the YZR, the palaeolakes located
on the west side of the YZR lacked a water supply, resulting in the disappearance of these
palaeolakes.

Data accessibility. The graphical data of Planation surface and valley shoulder elevation in different
parts of the middle reaches of YZR and the river network flow directions that supporting this paper
have been uploaded as electronic supplementary material.

Authors' contributions. Yong Liu, Liangshuai Wei, Tong Shen, Qinfeng Su, Anbang Huang and Yi Jia
conducted the experiments and data analysis, Yunsheng Wang guided the writing process and Yong
Liu wrote the paper. All the authors gave their final approval for publication.

Competing interests. The authors declare no competing interest.

Funding. This study was supported by the Foundation of China Geological Survey (DD20190325).

Acknowledgements. We thank the editor and anonymous reviewers for their valuable comments and
suggestions, and thank the American Journal Experts (AJE) for editing the grammar, phrasing, and
punctuation of the manuscript.

57 References

Brookfield M E. 1993. The Himalayan passive margin from Precambrian to Cretaceous sedimentary. *Geol.*, 84: 1-35.

Coleman M E, Parrish R R. 1995. Constraints on Miocene high-temperature deformation and anatexis within the Greater
Himalayan from U-Pb geochronology. *Earth Observation Satellite*, 76:708.
Chen J J, Ji J Q, Gong J F, et al. 2008. Formation of the Yarlung Zangbo Grand Canyon, Tibet, China. *Geological bulletin of*
*china*, 4:491-499.
Ding L, Cai F L, Zhang Q H, et al. 2009. Provenance and tectonic evolution of the foreland basin systems in the
Gandese-Himalayan collisional orogen belt. *Chinese Journal of Geology*, 4: 1289-1311.
Deng T, Hou S K, Wang N, et al. 2015. Hippurion fossils of the dati basin in Nyalam, Tibet, China and their
Paleocological and Paleoaltmetry implications. *Quaternary Sciences*, 3: 493-501.
Du B W, Peng Q H, Xie S K, et al. 2015. Exploration potential analysis of shale gas in the Lower Cretaceous,
Gamba-Tingri basin of Tibet. *Petroleum Geology and Recovery Efficiency*, 22(2): 51-54.
Ding L, Zhong D L. 1999. Characteristics and tectonics of high-pressure granulite facies metamorphism in Namjabawa
peak area, Tibet. *Science in China(Series D)*, 5: 385-397.
DAVID DIAN ZHANG. 1998. Geomorphological problems of the middle reaches of the Tsangpo River, Tibet. *Earth*
*Surface Processes and Landforms*, 23, 889-903.
Farr T G, Rosen P A, Caro E, et al. 2007. The Shuttle Radar Topography Mission. *Reviews of Geophysics*, 45(2): 1-33.
Gansser A. 1964. *Geology of the Himalayas*. New York: Interscience Publishers, 1-289.
Ge X H, Ren S M, Ma L X, et al. 2006. Multi-stage uplifts of the Qinghai-Tibet plateau and their environmental effects.
*Earth Science Frontiers*, 6: 118-130.
Harris N. 1995. Significance of weathering Himalayan metasedimentary rocks and leucogranites for the Sr isotope
evolution of sea water during early Miocene. *Geology*, 23(9): 759-798.
Han J E, Meng Q W, Guo C B, et al. 2017. Discovery and Significance of the Lacustrine Sedimentation in the Middle
Reach of Yarlung Zangbo River in Tibetan Plateau. *Geoscience—Journal of Graduate School, China University of Geosciences*,
5: 890-899.
Han J E, Yu J, Zhu D G, et al. 2011. Change of the paleoclimate and envoluyion of the lake during pliocene-early
pleistocene in zanda basin, Tibet. *Journal of Geomechanics*, 4: 361-372.
Laura B, Yani N, Randall R, et al. 2015. The Brahmaputra tale of tectonics and erosion: Early Miocene river capture in the
Eastern Himalaya. *Earth and Planetary Science Letters*, 415: 25-37.
Li L J. 1954. Natural area of Tibetan Plateau. *Journal of Geography*, 20(3): 255.
Liu S S, Yang Y F, Guo L N, et al. 2018. Tectonic characteristics and metallogeny in Southeast Asia. *Chinese Geology*, 5:
863-889.
Liu X F, Liu J Y, Zheng M P. 2007. Ostracods in the Jiu' er Paleolake, Zhongba, Tibet, Chian, and its environmentaland
climatic changes at 40000-10000 a BP. *Geological Bulletin OF China*, 1: 89-93.
Pan G T, Wang P S, Xu Y L, et al. 1990. *Cenozoic tectonic evolution of the Qinghai-Tibet Plateau*. Beijing: Geological
Press, 1-190.
Parrish R R, Hodges K V. 1993. Miocene (22±1 Ma) metamorphism and two stage thrusting in the Greater Himalayan
sequence, Annapurna Sanctuary, Nepal. *Geological Society of America Abstract with Program*, 25:174.
Pan Y S. 1999. Formation and uplift of the Qinghai-Tibet Plateau. *Geological Frontier*, 6(3): 153-163.
Qayyum M, Lawrence R D, Niem A R. 1997. Molasse-Delta-Flysch Continuum of the Himalayan Orogeny and Closure
of the Paleogene Katawaz Remnant Ocean, Pakistan. *International Geology Review*, 39(10): 861-875.
- Rea D K. 1992. Delivery of Himalayan sediment to the northern Indian Ocean and its relation to global climate, sea level
uplift, and seawater strontium. *Geophys. Monogr.*, 70: 387-402.
- Shi Y L, Li D W, Liu D M. 2010. Preliminary Study on the Inversion of Yalungzangbo River. *Geological Science and*
Technology Information, 29(4): 32-42.
- Saylor J E, Casturi L, Shanahan T M, et al. Tectonic and climate controls on Neogene environmental change in the Zhada

Basin, southwestern Tibetan Plateau. *Geology*, 44(1-11): 919-922.

Tapponnier P, Mattauer M, Proust F, et al. 1981. Mesozoic ophiolites, and large-scale tectonic movement in Afghanistan. *Earth and Planetary Science Letters*, 52:355-371.

Tachikawa T, Kaku M, Iwasaki A, et al. 2011. ASTER global digital elevation model version 2-summary of validation results. In: NASA.

Vance D, Bickle M, Ivy-Ochs S, et al. 2003. Erosion and exhumation in the Himalaya from cosmogenic isotope inventories of river sediments. *Earth and Planetary Science Letters*, 206: 273-288.

Wang E Q, Chen L Z, Chen Z L. 2002. Tectonic and climatic element – controlled evolution of the Yarlung Zangbo River in southern Tibet. *Quaternary Sciences*, 22(4): 365 – 373.

Wang F B, Shen X H. 1996. Formation and Evolution of the Kuala Lung Basin, Environmental Change and Himalayan Uplift. *Science in China(Series D)*, 4: 329-335.

Wang X X, Nie J S, Saylor J, et al. 2018. Environmental magnetic characteristics of Late Miocene fluvio-lacustrine sediments in Zhada basin and Indian monsoon evolution. *Quaternary Sciences*, 38(5): 1094-1100.

Xiang F, Wang C S, Zhu L D. 2002. Feature of cenozoic molasse at the south edge of Qinghai-Tibetan plateau. *Journal of Chengdu University of Technology*, 5: 515-519.

Yang Y C. 1982. The topographic features and the origin of the great bend valley at the Yarlung Zangbo River in Tibet. *Geographical Research*, 1(1):40-48.

Zhu D G, Meng X G, Shao Z G, et al. 2006. Quaternary glacial deposition and glacial advance and retreat in the Zanda basin and its surrounding mountains in Ngari, Tibet. *Chinese Geology*, 1: 86-97.

Zhu S, Zhao X T, Wu Z H. 2011. Response of Fluvial Landform of the Gyaca Sector of the Yarlung Zangbo River to Tectonic Movement and Climate. *Acta Geoscientica Sinica*, 3: 349-356.

Zhang P Q, Liu X H, Kong P. 2008. Evidence for glacial movement since last glacial period in the Great Canyon, Yarlung Zangbo, SE Tibet and its tectono–environmental implications. *Chinese Journal of Geology*, 3: 588-602.

Tables

Table 1 Information of the global DEMs and reference SRTM version.

SRTM version	Digital elevation data	Band	Special resolution	Central longitude (°)	Central latitude (°)	Horizontal/Vertical datum
SRTM3	srtm_53_06	C band	90 m	82.5	32.5	WGS84/EGM96
SRTM3	srtm_53_07	C band	90 m	82.5	27.5	WGS84/EGM96
SRTM3	srtm_54_06	C band	90 m	87.5	32.5	WGS84/EGM96
SRTM3	srtm_54_07	C band	90 m	87.5	27.5	WGS84/EGM96
SRTM3	srtm_55_06	C band	90 m	92.5	32.5	WGS84/EGM96
SRTM3	srtm_55_07	C band	90 m	92.5	27.5	WGS84/EGM96

* DEM data are from <http://www.gscloud.cn/>.

Table 2 Direction of the branches flowing into the YZR

River section	Riverbank	Direction of the branches flowing into the YZR		
		consequent	orthogonal	inverse
West of Jiacha Gorge	left bank	4	5	14
	right bank	0	0	8
East of Jiacha Gorge	left bank	6	0	2
	right bank	11	0	1

Figures

Fig. 1 Location of the Yarlung Zangbo River on the Tibet Plateau

Fig. 2 Flow chart of the river network extraction

Fig. 3 River network in the middle reaches of the YZR

Fig. 4 Directions of typical branches flowing into the YZR. A is Nianchu River, B is Lhasa River, C is Niyang River, all the rivers are the branches of YZR.

Fig. 5 Distribution and sedimentary characteristics of the palaeolakes

Fig. 6 Moraine terraces on the west side of Jiacha Gorge

Fig. 7 Tectonic geological map of the middle reaches of the YZR. A is the tectonic geological map from Qushui County to Jiacha County; B is the meandering river located at the lower reaches of the Jiacha Gorge outlet; C is the Quaternary sediments in Qusong Basin.

Fig. 8 Transverse section of the YZR west of Jiacha Gorge. P3 is the third-level planation surface, and V1 is the first-grade valley shoulder.

Fig. 9 Planation surface and valley shoulder elevation in different parts of the middle reaches of YZR. b-b' profile missing a first-grade valley shoulder on the left bank.

Fig. 10 Phase map of river terraces along the Jiacha sector of the YZR (modified after Zhu, 2011)

Fig. 11 Flow direction and fluvial sediments of the Menqu River

Fig. 12 Sedimentary characteristics of the branch fluvial sediments

Fig. 13 Flow direction and fluvial sediments of the Nianchu River. B is the profile of the Nianchu River Estuary.

Fig. 14 Evolution map of the YZR. Black arrow points to the direction of the plate movement.

Fig. 15 Topography of Qusong Basin and Qusong River

Fig. 16 Quaternary lacustrine sedimentary profile of Xialuo Township, Qusong County (modified from the 1:250000 regional geological survey)

Appendix B

Dear Editors and Reviewers,

Thank you for your comments on my manuscript. My responses to the comments are as follows:

Reviewer: 1

Comments to the Author(s)

This paper combines fieldwork and DEM analysis to understand the development of the understudied middle reaches of the Yarlung Zangbo River; it combines geomorphology and sedimentology.

Currently, the evidence for the analysis is missing for some sections, especially the branch deposits. A new section in the results section is required to show the fieldwork that you undertook. The implications of your work need to be included, how does this fit with the research further downstream?

Further analysis using the DEM could be used to strengthen your analysis, such as morphometric analysis of the river systems. Normalised steepness index and hypsometry can be used to show anomalous reaches of the river systems and can be completed using GIS. It would be interesting to see if there are any relationships between the branch river types and the main trunk river.

Response: A new section “5.2 Sedimentary characteristics of the tributary deposits” has been added to discuss the tributary deposits based on the fieldwork. This section is conducive to future research.

Specific comments below:

Page 1

Line 25 – Please expand on what you mean by Quaternary investigation – do you mean your field work? Or analysing the quaternary evolution of the catchment?

Response: I am sorry for this spelling, and the phrase “Quaternary investigation” means fieldwork. I have revised it in the revised version.

Line 28 – ‘it is believed’ – is this from previous published work? if so reword to ‘previous researchers have shown’

Response: I have reworded this sentence to “previous researchers have shown” in the revised version.

Line 34 – Water storage capacity is not the correct term, please look at the comments for page 6 line 11.

Response: I changed “Water storage capacity” to “total accommodation space” in the revised version.

Page 2

Section 3 – This section currently has a tectonic / structural background of the study area, geological information on the geology (bedrock and superficial including the stratigraphic successions of the palaeolake infill). Include a geological map / stratigraphic chart earlier than Figure 7. Geology is referred to throughout the paper, so setting the context here would be useful.

This section could also benefit from information on the types of river channels you are looking at – are they bedrock or bedrock-alluvial or alluvial in nature? What is the current catchment area of the river system?

Response: In the revised version, I have shown the types of river channels in the different river sections.

Line 51 – include the elevation decrease from east to west

Response: I have revised this sentence in the revised version.

Page 3

Line 57 – add what filling threshold you used

Response: The filling threshold used in the paper has been added.

Page 4

Line 5 – what area / con value did you use?

Response: The area used has been added to the revised version.

Section 5 – you refer to the deposits of the branch outlets in the analysis section (Section 6.1) – however do not present the data in the results sections from your fieldwork. What are the character of the deposits? Did you see any clast imbrication that supports flow reversal. A new sub-heading in the results section is needed to display your data to support your later interpretations.

Response: Data on the tributary outlet deposits have been added to section 5.2,

Page 4 /5

Section 5.2 – when each lake is introduced refer to the figure number (Figure 5), it would be useful to have a schematic diagram of the ages of the lakes and thickness of deposits – do you have a sediment logs of the area?

Response: The ages of the lake sediment were quoted from a previous study; no dating tests were conducted.

Page 6

Line 11 – Water storage capacity is not the right term here, sedimentary thicknesses can be high in areas of subsidence and low water volumes. Total accommodation space would be a more appropriate term to use.

Response: I changed “Water storage capacity” to “total accommodation space” in the revised version.

Line 22 / 23 – ‘Melting water from alpine glaciers...’ – I would remove this sentence, it doesn’t add anything.

Response: This sentence has been removed from the revised version.

Line 25 – 41 – are there any age constraints on these surfaces via dating e.g., cosmogenic or apatite fission track?

Response: The ages of the surfaces were quoted from a previous study.

Line 25 – 41 – the focus is on the c-c' cross sections, add in information about all the cross sections extracted (a-a' to e-e')

Response: Information on all the extracted cross-sections (a-a' to e-e') has been added to the revised version.

Line 43 – 58 – references on bedrock channels are missing from this section.

For example, but not limited to:

Tinkler, K.J. and Wohl, E., 1998. Rivers over rock: Fluvial processes in bedrock channels (No. 107). American Geophysical Union. And references therein

Richardson, K. and Carling, P., 2005. A typology of sculpted forms in open bedrock channels (Vol. 392). Geological Society of America.

Response: References on bedrock channels have been added to the revised version.

Line 44 – Reword 'large drop' to 'steep gradient' and include the gradient of the reach

Response: I have revised it in the revised version.

Line 48 – Reword 'There are no quaternary deposits in the gorge due to the strong erosional downcutting of the river...' In a bedrock setting, deposits are not expected in the reach due to the high stream powers causing high transport capacity of the water.

Response: The sentence has been revised in the revised version.

Page 7

Section 6 in general– what are the implications of your study? Do any of the dates correspond with information in the depositional basin? Does this support previously published research on the lower reaches or not?

Response: I apologize for the lack of clarity in this section, and I have added some information about the Jiahca Gorge and the palaeolakes to support the arguments.

Section 6.1 – the evidence is needed in the results section to support your interpretations for this section.

Response: I have added some information on the tributaries of the YZR in the results to support the interpretations.

Section 6.2 – how does your results and interpretation fit with the other work further downstream cited in the introduction?

Response: I have revised this section, and we combine our work with previous research and results for interpretation in this section.

Page 8

Line 5 – do you have a reference for the elevation of the plateau?

Response: The reference for the elevation of the plateau has been added.

Line 17 – large water capacity, see comment for page 6 line 11

Response: This sentence has been revised.

Page 9

Line 14 – ‘were burst by the river’s upstream development’ reword to ‘drained by headward erosion’

Response: This sentence has been reworded to “drained by headward erosion”.

Line 31 – water storage capacities, see comment for page 6, line 11.

Response: This sentence has been revised.

Figures

Figure 1 – The text in the inset box should be increased, when printed, this is hard to read, a legend is needed for the DEM in the inset box.

What do the yellow dashed line refer to? Missing from the key

Response: Thank you for your advice. I have revised these issues in the revised version.

Figure 2 – it would be useful to include information on the thresholds you used

Response: Information on the thresholds has been added.

Figure 3 – Move the inset b, below part A – it is currently too small to see and could be made much larger if you place it below A

Response: Thank you for your advice. I have revised these issues in the revised version.

Figure 4 – place a white box behind the scale, and change the text to black / bold the legend – it is currently hard to see

Response: I have revised these issues in the revised version.

Figure 5 – you could add the sediment logs in this figure for B, C and D (See comment for section 5.2)

Response: Thank you for your advice. Sediment logs have been added.

Figure 7 – the red box in inset A is hard to see with the current colour scheme used

Response: The red box in inset A has been revised.

Figure 8 – are the moraines mentioned in section 5.3 visible in any of the cross sections?

Response: The moraine terraces are located on the left bank of the Woka River, but no cross-section passes through the moraine terraces.

Figure 14 – in the text you mention about draining of the lakes by headward erosion / shrinkage of the lakes, it would be good to show this in the schematic as section C

Response: This section has been revised.

Figure 15 – reword caption to ‘Satellite image of the Qusong Drainage Basin’ – or add a scale for the elevation

Response: This sentence has been removed.

Reviewer: 2

Comments to the Author(s)

The authors presented many figures which can be merge and decrease the numbers of figures. I made some corrections but I found the lack of writing flow so suggest to improve English writing and re-phrase most of the sentences. Although the result is excellent but due to the lack of writing flow it is hard for readers to get into it.

I suggest the paper for major revision with English re-editing.

Response: Thank you for your advice. The figures have been merged and revised, and all the sentences have been rephrased with the help of AJE.

Appendix C

Dear Editors and Reviewers,

Thank you for so much for your comments and accepting for my manuscript. My responds to the comments are as follows:

Comments to the Author(s)

The manuscript has improved and the figures are easier to see, the addition of the new section 5.2 is welcomed. However, in its current form it is not sufficient to back up your analysis. There is no actual data presented in your paper - sedimentary logs, photos of the deposit, imbrication data. In the previous version 'Figure 5' could have been expanded to add this - which has now been removed. Without the actual evidence, it is hard to accept the descriptions now added in Section 5.2.

The interplay between climate and tectonics is also really interesting, and could be expanded on to show wider global implications.

Specific comments below:

Section 5.2 – Geology information is needed to set the context, currently this just looks at the geomorphic setting of river types and the tectonic setting. Figure 4 includes information on geology (rock types) –add some text about the geology here.

Response: Some information on geology of figure 4 has been added in this section.

Page 9, line 7 – reword 'the river bed is composed of bedrock due to strong erosion' to 'the bedrock gorges are areas of high erosion rates and stream power'

Response: I have reworded 'the river bed is composed of bedrock due to strong erosion' to 'the bedrock gorges are areas of high erosion rates and stream power'.

Page 9, line 9 – reword largest to deepest gorge

Response: In the manuscript, all the “largest gorge” have been reworded to “deepest gorge”.

Page 11, line 10 – change 'extrusion' to 'convergence'

Response: The 'extrusion' has been changed to 'convergence'.

Page 11, line 25 – just put '87%', the decimal place isn't needed

Response: Thank you for your advice, and the “87.1%” has been reworded to “87%”.

Page 12, line 39 to 41 – 'this finding further indicates....' – this is interpretation ad shouldn't be in the results section

Response: This sentence has been removed.

Section 5.2 – More evidence is needed in this section of your fieldwork - where is the evidence / data of imbrication? Did you take a picture? What was the strike/dip? how many clasts did you measure, can you create a stereonet?

A sedimentary log is needed with pictures of the different facies types (e.g., sand lenses, imbrications)

Do you have other clast data – lithology?

Response: One picture of the sediments and some information about the lenses and imbrications

have been added in this section.

Page 14, line 34 – it would be good to bring the imbrication in here

Response: Some information about the imbrication has been added in here.

Page 15, line 31 – reword largest to deepest

Response: This word has been reworded.

Page 15, line 37 – reword continental river to ‘endorheic basin’

Response: This sentence has been reworded.

Page 16 – there is an interesting interplay between tectonics and climate influencing the basin, a paragraph should be added to highlight this – this will show wider implications of the study

Conclusions – It would be good to say how this relates to the other research and bring back the information cited in the introduction. Even if the sentence is to say, how your work has added to the story ‘this research has confirmed the work downstream (REFS)’ for clarity.

Response: A paragraph on the influence of tectonics and climate has been added in this section, and the conclusions have been revised according to the advice.

Figure 1 – the DEM legend needs units (m)

Response: The DEM legend of figure 1 has added units.

Figure 2 – you could increase the size of the D and E insets to make them bigger and use all of the white space

Response: The insets of D and E has been increased.

Figure 7 - the white text is still hard to read, either have a box behind the text (white box, black text) or change the text to a different colour and bold it

Response: A white box behind the text has been added.

Figure 8 – pop a white box behind the scale

Response: A white box behind the scale has been added.

Figure 10 – Can you add any rough time indications of the evolution? - annotate the stream capture on E. The 'Jiacha mountain' is not referred to in the text - it is currently hard to place C - E in the context of A and B.

Response: The Jiacha gorge is located in the Jiacha mountain, some sentences have been revised.

All revision in this manuscript has been marked in red.